# Robust Linear Dueling Bandits with Post-serving Context under Unknown Delays and Adversarial Corruptions

Youngmin Oh [1]

## Abstract

We study linear dueling bandits in volatile environments characterized by the simultaneous presence of post-serving contexts, delayed feedback, and adversarial corruption. Feedback is subject to unknown stochastic or adversarial delays and a cumulative corruption budget $\mathcal{C}$. To address these challenges, we propose RCDP-UCB, which integrates a learned approximator that predicts post-serving contexts from pre-serving information. It further employs an adaptive weighting strategy that clips feature vectors to mitigate the impact of corrupted and delayed observations simultaneously. Under standard regularity conditions and a parametric post-serving mapping, we rigorously establish that our algorithm is delay-regime-agnostic, achieving a regret upper bound of $\widetilde{\mathcal{O}}(d(\sqrt{T}+\mathcal{C}+\mathcal{D}))$, where $d$ is the total feature dimension and $\mathcal{D}$ encapsulates the delay complexity, scaling with $\sqrt{\Lambda}$ under adversarial delays or $\mu_\tau$ under stochastic delays ($\Lambda$: cumulative delay budget; $\mu_\tau$: mean of sub-Gaussian delays). We further establish lower bounds that nearly match our upper bounds up to a $\sqrt{d}$ factor for adversarial delays in the absence of post-serving contexts. Code is available at https://github.com/youngmin0oh/rcdp-public.

## 1. Introduction

While the multi-armed bandit (MAB) framework successfully models various decision-making problems—ranging from recommendation systems to ad allocation (Lattimore & Szepesvári, 2020)—designing explicit real-valued reward functions often proves elusive in practice. Indeed, in many modern interactive systems, user feedback is inherently relative rather than absolute. For instance, a user may readily prefer Movie A over Movie B, yet find it difficult to assign a precise numerical score to either.

This reliance on relative feedback has become increasingly critical with the advent of Large Language Models (LLMs) (Guo et al., 2025). Reinforcement Learning from Human Feedback (RLHF) (Ouyang et al., 2022; Rafailov et al., 2023), one of the techniques for training LLMs, fundamentally relies on pairwise preference comparisons to align model outputs with human intent, demonstrating the pivotal role of preference-based learning in state-of-the-art AI systems (Guo et al., 2025; Achiam et al., 2023). This preference-based learning is elegantly formulated by the dueling bandits framework (Yue et al., 2012). To leverage side information for personalization in these complex environments, this framework has been further extended to the contextual setting (Dudík et al., 2015; Saha, 2021), bridging the gap between theoretical preference modeling and practical, high-dimensional applications.

However, existing contextual dueling bandit algorithms predominantly operate under the assumption that the utility of an arm is fully determined by the pre-serving context—features observable before an action is selected (e.g., user demographics, item metadata). This assumption fails to capture critical factors that are only revealed after the service is rendered, which we term post-serving contexts following Wang et al. (2024). For instance, in food delivery, the true enjoyment of a meal depends not only on the restaurant's cuisine (pre-serving) but also on the actual delivery time and food temperature (post-serving). Similarly, in ride-sharing, a user's satisfaction depends on the car type (pre-serving) as well as the cleanliness and driver conduct (post-serving). Ignoring these latent, post-hoc determinants can lead to suboptimal policy learning, as the learner fails to account for the full causal mechanism driving user preferences.

Compounding this challenge, real-world feedback is rarely instantaneous or pristine. Feedback often arrives with unknown stochastic or adversarial delays (Vernade et al., 2020; Lancewicki et al., 2023; Howson et al., 2023), and the observed outcomes may be subject to malicious corruption (Liu et al., 2024; Di et al., 2025). The interplay between latent post-serving contexts, delayed feedback, and

[1]InfiniTree, Republic of Korea. Correspondence to: Youngmin Oh <youngmin0.oh@gmail.com>.

*Proceedings of the 43rd International Conference on Machine Learning*, Seoul, South Korea. PMLR 306, 2026. Copyright 2026 by the author(s).

*Table 1.* Comparison of cumulative regret upper bounds for various contextual bandit settings. We further omit sub-leading terms to highlight the dominant scaling (e.g., $\lambda$).

| SETTING | CORRUPTION | DELAY | ALGORITHM | UPPER BOUND |
|---|---|---|---|---|
| LINEAR BANDITS | NO | NO | OFUL (ABBASI-YADKORI ET AL., 2011) | $\widetilde{\mathcal{O}}(d\sqrt{T})$ |
| | YES | NO | CW-OFUL (HE ET AL., 2022) | $\widetilde{\mathcal{O}}(d\sqrt{T} + d\mathcal{C})$ |
| | NO | STOCHASTIC | DELAYED-UCB (ZHOU ET AL., 2019) | $\widetilde{\mathcal{O}}(\sqrt{dT(d + \mathbb{E}[\tau])})$ |
| | | | OTFLINUCB* (VERNADE ET AL., 2020) | $\widetilde{\mathcal{O}}((d\sqrt{T} + md)/\tau_m)$ |
| | | | GLB-UCB† (HOWSON ET AL., 2023) | $\widetilde{\mathcal{O}}(d\sqrt{T} + d^{3/2}\mathbb{E}[\tau])$ |
| | NO | ADVERSARIAL | ITO ET AL. (2020) | $\widetilde{\mathcal{O}}(\sqrt{d(d + d_{\max})T})$ |
| GENERALIZED LINEAR BANDITS | NO | NO | GLM-UCB (FILIPPI ET AL., 2010) | $\widetilde{\mathcal{O}}(d\sqrt{T})$ |
| | YES | NO | GADAOFUL (YU ET AL., 2025) | $\widetilde{\mathcal{O}}(d\sqrt{\sum_t \sigma_t^2} + d\mathcal{C})$ |
| | YES | NO | YE ET AL. (2023) | $\widetilde{\mathcal{O}}(d\sqrt{T} + d\mathcal{C})$ |
| | NO | STOCHASTIC | DELAYED-UCB (ZHOU ET AL., 2019) | $\widetilde{\mathcal{O}}(d\sqrt{T} + d\mathbb{E}[\tau]\sqrt{T})$ |
| | | | GLB-UCB (HOWSON ET AL., 2023) | $\widetilde{\mathcal{O}}(d\sqrt{T} + d^{3/2}\mathbb{E}[\tau])$ |
| LINEAR DUELING BANDITS | NO | NO | SAHA (2021); BENGS ET AL. (2022) | $\widetilde{\mathcal{O}}(d\sqrt{T})$ |
| | YES | NO | RCDB (DI ET AL., 2025) | $\widetilde{\mathcal{O}}(d\sqrt{T} + d\mathcal{C})$ |
| | YES | BOTH | RCDP-UCB (OURS) | $\widetilde{\mathcal{O}}(d(\sqrt{T} + \mathcal{C} + \mathcal{D}))$ |

$d$: feature dimension; $T$: time horizon; $\mathcal{C}$: total corruption budget; $\mathbb{E}[\tau]$: expected delay; $d_{\max}$: maximum delay; $\kappa$: lower bound of the link function derivative; $\sigma_t^2$: variance at round $t$; $\mathcal{D} = \max(\mu_\tau, \Lambda^{1/2})$, where $\mu_\tau$ is the mean of the sub-Gaussian delay and $\Lambda$ is the total adversarial delay budget; $\tau_m = \mathbb{P}(D \le m)$ denotes the cumulative probability of the delay falling within window $m$.

adversarial corruption creates a hostile environment where standard regret minimization strategies typically fail.

Specifically, the uncertainty in mapping pre-serving to post-serving contexts introduces non-stationarity, while delays and corruption can severely skew the learner's perception of utility. Solving this tripartite problem poses a substantial theoretical challenge, as the combined effect of these factors is fundamentally multiplicative rather than additive. Delays significantly reduce the effective sample size available for immediate updates, creating a data-scarce regime. This sparsity amplifies the statistical influence of adversarial corruptions: in the absence of abundant data, each observed feedback signal carries disproportionately high leverage. Consequently, an adversary can exert considerable bias by selectively corrupting these sparse observations, rendering the learning process highly sensitive to minor perturbations. Moreover, the learner must estimate a mapping from pre-serving to post-serving contexts using already-compromised and delayed feedback. This creates a vicious cycle: corrupted signals degrade the context predictor, which in turn magnifies estimation errors where delays have already starved the model of reliable data.

To surmount these intertwined challenges, we propose RCDP-UCB (Robust to Corruption, Delay, and Post-serving UCB), an algorithmic framework. To handle corrupted and delayed observations, we employ a weighted ridge regression estimator with an uncertainty-aware re-weighting

mechanism. The intuition is as follows: observations that appear unreliable—either due to excessive delay or potential corruption—receive lower weight. This allows the learner to gracefully degrade under attack rather than catastrophically fail. In our setting, post-serving contexts are revealed after the decision, forcing the learner to act under partial information. We construct a confidence ellipsoid that explicitly accounts for this additional variance, enabling principled optimism over both the unknown preference parameters and the unobserved features.

A key feature of our approach is that it does not require prior knowledge of the specific delay mechanism type (stochastic vs. adversarial). While the aggregate magnitude of the corruption and delay budgets is utilized for theoretical tuning of the clipping threshold $\alpha$, the algorithm adapts efficiently without needing to detect or classify the source of the delay.

We establish the first regret guarantees for linear dueling bandits within this unified tripartite setting. Under standard regularity conditions and a parametric post-serving mapping, we achieve a regret bound of: $\widetilde{\mathcal{O}}\left(d\left(\sqrt{T} + \mathcal{C} + \mathcal{D}\right)\right)$, where $d$ denotes the feature dimension, $T$ is the time horizon, and $\mathcal{C}$ represents the cumulative corruption budget. Notably, our framework provides best-of-both-worlds guarantees in terms of delay regimes for the delay complexity $\mathcal{D}$, which adaptively scales as $\Lambda^{1/2}$ for an adversarial delay budget $\Lambda$, or as the mean delay $\mu_\tau$ for sub-Gaussian distributions. Each component of the bound admits a transparent interpre-

tation: $d\sqrt{T}$ represents the fundamental cost of learning, $d\mathcal{D}$ captures the penalty incurred by delayed feedback, and $d\mathcal{C}$ reflects the impact of adversarial corruptions.

To certify the optimality of our results, we derive a lower bound of $\Omega\left(\sqrt{dT} + d\mathcal{C} + \mathcal{D}'\right)$ in the absence of post-serving contexts, where $\mathcal{D}' = \max\{\sqrt{d\Lambda}, d\mu_\tau\}$. This confirms that the overhead introduced by delays and corruptions is information-theoretically unavoidable, and that RCDP-UCB achieves this limit with near-optimal efficiency up to a $\sqrt{d}$ factor for adversarial delays. To the best of our knowledge, this work is the first to simultaneously address the joint challenges of adversarial corruptions and delayed feedback within the contextual dueling bandit framework.

Our primary contributions are summarized as follows:

- **Unified Analytical Framework:** We formalize a linear dueling bandit framework tailored for volatile environments, providing a comprehensive treatment of three critical practical challenges: post-serving contexts, unknown observation delays, and adversarial corruptions.

- **Algorithm Design:** We propose RCDP-UCB, a novel algorithm that integrates contextual mapping estimation with adaptive clipping mechanisms by re-weighting. This re-weighting strategy simultaneously controls both corrupted and delayed observations, enabling the algorithm to remain agnostic to whether feedback irregularities stem from adversarial manipulation or network latency.

- **Regret Analysis:** We prove a regret upper bound of $\widetilde{\mathcal{O}}(d(\sqrt{T} + \mathcal{C} + \mathcal{D}))$, matching our lower bound up to a $\sqrt{d}$ factor in the adversarial delay regime.

## 2. Related Work

**Linear Contextual Dueling Bandits.** Linear utility functions in CDBs have been extensively studied (Bengs et al., 2022; Saha, 2021; Di et al., 2023; Li et al., 2024). Bengs et al. (2022) and Saha (2021) established UCB-based regret bounds of $\widetilde{\mathcal{O}}(d\sqrt{T})$ and $\widetilde{\mathcal{O}}(\sqrt{dT})$, matching the lower bounds identified by Saha (2021). Alternative approaches include the variance-aware action-elimination method by Di et al. (2023) and the Thompson Sampling-based algorithm by Li et al. (2024), which also achieves $\widetilde{\mathcal{O}}(d\sqrt{T})$ regret. Beyond the linear regime, Oh et al. and Verma et al. (2025) have extended CDBs to neural function classes via deep feature representation and NTK approximations, providing UCB- and TS-based exploration strategies with provable regret guarantees.

**Robustness to Adversarial Corruption.** Addressing the vulnerability of bandit algorithms to data poisoning remains a pivotal challenge. A fundamental distinction exists between action-independent (weak) and action-dependent (strong) corruption. In the context of linear bandits, He et al. (2022) and Liu et al. (2024) established the minimax regret bounds under corruption. For Generalized Linear Bandits (GLBs), Yu et al. (2025) introduced GAdaOFUL, a variance-aware algorithm utilizing adaptive Huber regression. More recently, Ye et al. (2023) proposed CR-GLM-UW, which leverages uncertainty weighting to achieve robust performance without prior knowledge of the corruption level. In the realm of Dueling Bandits, while earlier works like Agarwal et al. (2021) addressed general robustness, Di et al. (2025) recently developed RCDB, achieving nearly optimal regret bounds in the presence of strong corruptions.

**Delayed Feedback Mechanisms.** Delays in outcome observation significantly complicate the regret landscape. For stochastic delays in linear bandits, Zhou et al. (2019) provided foundational upper bounds, which Vernade et al. (2020) extended to scenarios with censored feedback (OTFLinUCB). In the GLB setting, Howson et al. (2023) demonstrated that with appropriate optimism, the regret penalty remains additive with respect to the expected delay. Regarding adversarial delays, Ito et al. (2020) derived regret bounds dependent on the maximum delay $d_{\max}$. Despite these advances, the intersection of heavy-tailed delays and adversarial corruption—particularly in the preference-based setting—remains largely unexplored, a gap our work aims to bridge.

**Post-Serving Contexts and Partial Observability.** When valuable contextual information is only observable after arm selection, standard contextual bandit algorithms may suffer from model misspecification. Wang et al. (2024) introduced the framework of contextual bandits with post-serving contexts, proposing the poLinUCB algorithm that achieves regret $\widetilde{\mathcal{O}}(T^{1-\alpha}d_u^\alpha + d_u\sqrt{TK})$, where $\alpha \in [0, 1/2]$ captures the learnability of the pre- to post-context mapping function. Central to their analysis is a robustified Elliptical Potential Lemma (EPL) that accommodates noise in observed features. Earlier work by Wang et al. (2016) studied hidden contexts under strong initialization assumptions, while subsequent research explored noisy or unobservable contexts through online prediction from context histories (Qi et al., 2018; Yang et al., 2020; Yang & Ren, 2021; Zhu & Kveton, 2022). Park & Faradonbeh (2021) further analyzed the Thompson Sampling algorithm under partial observability constraints.

# 3. Problem Formulation

**Basic Setup with Post-serving Contexts.** We consider a contextual dueling bandit problem over a finite horizon $T$ with a finite $K$ arms. In each round $t \in [T]$, the environment provides a pre-serving context set $\mathcal{X}_t = \{x_{t,1}, \ldots, x_{t,K}\} \subset \mathbb{R}^{d_x}$. Then the learner selects a pair of arms $(a_t, b_t) \in [K] \times [K]$. Following the selection, the learner observes the associated post-serving contexts $y_{t,a_t}, y_{t,b_t} \in \mathbb{R}^{d_y}$ as in Wang et al. (2024). The complete feature representation for an arm $k$ is defined as the concatenation $z_{t,k} = (x_{t,k}, y_{t,k}) \in \mathbb{R}^d$, where $d = d_x + d_y$. We assume that there is a learnable mapping $\phi_*(\cdot) : \mathbb{R}^{d_x} \to \mathbb{R}^{d_y}$ such that $y_{t,k} = \phi_*(x_{t,k}) + \epsilon_{t,k}$ where $\epsilon_{t,k}$ is a zero-mean noise vector, i.e., $\phi_*(x_{t,k}) = \mathbb{E}[y_{t,k}|x_{t,k}]$. We adopt a linear utility model with an unknown parameter $\Theta_* \in \mathbb{R}^d$, where the utility of arm $k$ at round $t$ is given by $u_{t,k} = \langle \Theta_*, z_{t,k} \rangle$ for $k \in [K]$. The preference probability between arms $a_t$ and $b_t$ is governed by a link function $g : \mathbb{R} \to [0,1]$. Specifically, the binary outcome $l_t \in \{0,1\}$ satisfies $\mathbb{E}[l_t \mid a_t, b_t] = g(\langle \Theta_*, z_{t,a_t} - z_{t,b_t} \rangle)$. A prominent example is the logistic link function $g(x) = \frac{1}{1+\exp(-x)}$, corresponding to the Bradley-Terry-Luce (BTL) model (Hunter, 2004; Luce, 2005).

**Feedback Delay.** A distinguishing feature of our setting is that the learner operates without prior knowledge of whether the delay sequence $\{\tau_t\}_{t=1}^T$ is governed by a stochastic process or determined by an adversary. The delay regime is fixed throughout the entire horizon but remains unknown to the learner. We consider two canonical settings:

1. Stochastic Delays. The delays $\{\tau_t\}_{t=1}^T$ are independent $\sigma^2$-sub-Gaussian random variables with mean $\mu_\tau$. That is, for all $n \in \mathbb{R}$, $\mathbb{E}\big[\exp\big(n(\tau_t - \mu_\tau)\big)\big] \leq \exp\left(\frac{n^2\sigma^2}{2}\right)$ as in Howson et al. (2023).

2. Adversarial Delays. An adaptive adversary selects the delay sequence subject to a cumulative budget constraint: $\sum_{t=1}^T \tau_t \leq \Lambda$. The adversary may choose $\tau_t$ based on $\mathcal{H}_{t-1}$, consisting of all rounds whose feedback has arrived (formally defined below).

We define the *observed-feedback history* at round $t$ as

$$\mathcal{H}_t := \big\{ s \in [t] : s + \tau_s \leq t \big\},$$

i.e., the set of past rounds whose (possibly corrupted) outcome $o_s$ has been revealed to the learner by the end of round $t$. The mean $\mu_\tau$ is meaningful only in the stochastic-delay regime, while the budget $\Lambda$ applies only in the adversarial-delay regime; throughout the paper, $\mu_\tau$ and $\Lambda$ are used solely in their respective regimes.

**Feedback Corruption.** Before the outcome is revealed to the learner, an adversary may corrupt the true signal $l_t$ to produce a corrupted observation $\gamma_t \in \{0,1\}$. The adversary is constrained by a total corruption budget $C$: $\sum_{t=1}^T |l_t - \gamma_t| \leq C$. Accordingly, we let $o_t$ be the observed outcome to the learner, which is unknown whether $o_t = l_t$ or not.

**Regret Definition.** The learner's objective is to minimize the cumulative average regret $R_T$. Let $k_t^* = \arg\max_{k \in [K]}\langle \Theta_*, z_{t,k}^* \rangle$ be the optimal arm at round $t$, where $z_{t,k}^* = (x_{t,k}, \phi_*(x_{t,k})) \in \mathbb{R}^d$. The instantaneous regret $r_t$ is defined as the utility gap between the optimal arm and the selected pair (Saha, 2021):

$$r_t = \frac{1}{2}\left( \langle \Theta_*, z_{t,k_t^*}^* - z_{t,a_t}^* \rangle + \langle \Theta_*, z_{t,k_t^*}^* - z_{t,b_t}^* \rangle \right).$$

The total cumulative regret is thus $R_T = \sum_{t=1}^T r_t$. The learner must achieve sublinear regret while remaining agnostic to the specific delay regime and the presence of corruption.

# 4. RCDP-UCB

To establish the theoretical guarantees for our proposed algorithm, we introduce the following structural and statistical assumptions. These are standard in the literature on generalized linear bandits and contextual dueling bandits with auxiliary information (Pike-Burke et al., 2018; Di et al., 2025; Bengs et al., 2022; Di et al., 2023).

**Assumption 4.1.** The link function $g : \mathbb{R} \to [0,1]$ is continuously differentiable, and there exists a constant $\kappa > 0$ such that its derivative satisfies $\dot{g}(s) \geq \kappa$ for all $s$ in the domain of interest.

**Assumption 4.2.** The true underlying parameter $\Theta_* \in \mathbb{R}^d$ is contained within a ball of radius $M$, i.e., $\|\Theta_*\|_2 \leq M$ for some known constant $M > 0$.

**Assumption 4.3** ($\mathcal{L}_2$-Consistency and Convergence Rate). Let $\mathcal{D}$ be a probability distribution over $\mathcal{X} \times \mathcal{Y}$. We assume the existence of an estimation algorithm that, given a dataset $S = \{(x_s, y_s)\}_{s=1}^t$ sampled i.i.d. from $\mathcal{D}$, produces an estimator $\widehat{\phi}_t : \mathcal{X} \to \mathbb{R}^{d_y}$. Furthermore, we assume that there exist constants $C_0 > 0$ and $a \in (0, 1/2]$ such that for any $\delta \in (0,1)$, the estimator $\widehat{\phi}_t$ satisfies the following uniform error bound with probability at least $1 - \delta$:

$$\sup_{x \in \mathcal{X}} \|\widehat{\phi}_t(x) - \phi_*(x)\|_2 \leq C_0 \frac{\sqrt{d_x}}{t^a} \log(t/\delta),$$

where $\phi_*$ is the underlying ground-truth function.

*Remark* 4.1 (Spectral and Smoothness Interpretations). Assumption 4.3 encapsulates various functional regimes:

- Parametric Linear Case: If $\phi_*(x) = \Phi^\top x$, then $a = 1/2$ is achievable via ordinary least squares or ridge regression (Abbasi-Yadkori et al., 2011).

- Non-parametric Hölder Continuity: For $\phi_* \in \mathcal{H}_\beta(\mathcal{X})$, the minimax optimal rate is $a = \beta/(2\beta + d_x)$, reflecting the curse of dimensionality (Tsybakov, 2008).

- Reproducing Kernel Hilbert Spaces (RKHS): For functions with bounded norm in an RKHS, $a = 1/2$ can be recovered under standard regularity conditions (Srinivas et al., 2012).

We assume the boundedness of contexts in the $L_2$ norm for the sake of simplicity:

**Assumption 4.4.** The joint feature vectors are bounded within a ball of radius $1/2$, i.e., $\|z_{t,k}\|_2 \le 1/2$ for all $t \in [T]$ and $k \in [K]$. Consequently, the pairwise feature differences satisfy $\|\Delta z_{t,k_1,k_2}\|_2 \le 1$, where $\Delta z_{t,k_1,k_2} := z_{t,k_1} - z_{t,k_2}$.

**Assumption 4.5** (Delay Regime)**.** The delay sequence $\{\tau_t\}_{t=1}^T$ follows exactly one of the two regimes introduced in Section 3: (i) *stochastic delays* — the $\tau_t$ are independent $\sigma^2$-sub-Gaussian random variables with mean $\mu_\tau$; or (ii) *adversarial delays* — the $\tau_t$ are chosen by an adaptive adversary subject to $\sum_{t=1}^T \tau_t \le \Lambda$. The mean $\mu_\tau$ is used only in regime (i) and the budget $\Lambda$ only in regime (ii); the algorithm is agnostic to which regime is in effect.

We define the following weighted design matrix to facilitate our analysis with weights $\{\omega_s\}_{s=1}^t$:

$$\widetilde{V}_t = \lambda I + \kappa \sum_{s=1}^t \omega_s \Delta z_s \Delta z_s^\top, \tag{1}$$

$$\widetilde{W}_t = \lambda I + \kappa \sum_{s=1}^t \mathbb{I}\{s + \tau_s \le t\} \omega_s \Delta z_s \Delta z_s^\top, \tag{2}$$

with $\Delta z_s = \Delta z_{s,a_s,b_s}$, where $a_s$ and $b_s$ are chosen arms at round $s$. We estimate the unknown parameter $\Theta_* \in \mathbb{R}^{d_x + d_y}$ by minimizing the weighted regularized negative log-likelihood, which corresponds to the Maximum Likelihood Estimation (MLE) with an $\ell_2$-regularizer:

$$\mathcal{L}_t(\Theta) = \frac{\lambda}{2}\|\Theta\|_2^2 - \sum_{s \in \mathcal{H}_{t-1}} \omega_s \log g\left((-1)^{1-o_s}\langle \Theta, \Delta z_s \rangle\right),$$

where $\Delta z_s = \Delta z_{s,a_s,b_s}$, and $\omega_s > 0$ is the weight for sample $s$. The estimator $\Theta_t$ is obtained as:

$$\Theta_t = \underset{\Theta \in \mathbb{R}^{d_x + d_y}}{\arg\min} \mathcal{L}_t(\Theta). \tag{3}$$

Assuming $g$ belongs to a set of exponential family distributions, the following estimating equation is obtained (Bengs et al., 2022):

$$\lambda \Theta_t + \sum_{s \in \mathcal{H}_{t-1}} \omega_s \left(g(\Theta_t^\top \Delta z_s) - o_s\right) \Delta z_s = 0.$$

To simultaneously mitigate the impact of adversarial corruption and stabilize the bias induced by delayed feedback, we propose the following adaptive weighting scheme. Let $\alpha > 0$ serve as a tuning parameter for robustness. We define the weight $\omega_s$ for the $s$-th sample as:

$$\omega_s = \min\left(1, \frac{\alpha}{\|\Delta z_s\|_{\widetilde{V}_{s-1}^{-1}}}\right). \tag{4}$$

Geometrically, this weighting enforces the soft constraint $\|\sqrt{\omega_s}\Delta z_s\|_{\widetilde{V}_{s-1}^{-1}} \le \alpha$. It serves as a unified mechanism mitigating both adversarial corruptions and delay-induced bias. Note that while Di et al. (2025) utilized this strategy exclusively for robustness against corruption, our work establishes it as a unified mechanism that simultaneously counteracts both adversarial corruption and delayed feedback, with $\alpha = \sqrt{d}/(\mathcal{C} + \mathcal{D})$ tuned to achieve sublinear regret.

Furthermore, we let $\widehat{\phi}_t$ be an estimator of $\phi_*$ at round $t$. Then we propose the following strategy RCDP-UCB: at each round $t$,

$$a_t = \underset{k \in [K]}{\arg\max} \langle \Theta_{t-1}, \widehat{z}_{t,k} \rangle, \tag{5}$$

$$b_t = \underset{k \in [K]}{\arg\max} \langle \Theta_{t-1}, \widehat{z}_{t,k} \rangle + c_t \|\Delta \widehat{z}_{t,k,a_t}\|_{\widetilde{V}_{t-1}^{-1}}. \tag{6}$$

where $\widehat{z}_{t,k} = (x_{t,k}, \widehat{y}_{t,k})$ with $\widehat{y}_{t,k} = \widehat{\phi}_t(x_{t,k})$ and $\Delta z_{t,k,k'} := \widehat{z}_{t,k} - \widehat{z}_{t,k'}$ for $k, k' \in [K]$. Note that a learner cannot observe the post-serving contexts, so the learner estimates $\widehat{y}_{t,k}$ of $y_{t,k}$ by utilizing $\widehat{\phi}_t$. The value $c_t$ will be specified later when analyzing the regret analysis. The full procedure is summarized in Algorithm 1.

Our algorithm, RCDP-UCB, employs $\widetilde{V}_t$ for both arm selection and robust weight calculation, rather than the observed matrix $\widetilde{W}_t$. This approach enforces optimism in the face of scheduled uncertainty, effectively preventing redundant exploration in delayed settings and ensuring the selection strategy is consistent with the robustness mechanism. While exploration and weighting leverage this optimistic $\widetilde{V}_t$, parameter estimation ($\Theta_t$) remains strictly grounded in realized feedback via $\widetilde{W}_t$, ensuring statistical validity.

## 5. Regret Analysis

A fundamental theoretical bottleneck in this regime arises from the multiplicative interplay between feedback latency and adversarial corruption, a challenge further exacerbated by post-serving contexts. Conventional analytical frameworks typically decompose the weighted norm associated with the observed Gram matrix $\widetilde{W}_t$ as follows $\|\Delta z_s\|_{\widetilde{W}_t^{-1}} \lesssim \|\Delta z_s\|_{\widetilde{V}_t^{-1}} + \|\Delta z_s\|_{\widetilde{M}_t^{-1}}$, where $\widetilde{M}_t$ represents the adjustment matrix accounting for delayed observations. Following Howson et al. (2023), the delay adjustment satisfies

---

**Algorithm 1** RCDP-UCB

---

**Input:** Horizon $T$, dimension $d = d_x + d_y$, regularization $\lambda$, robustness parameter $\alpha$, design matrix $\widetilde{V}_0 = \lambda I$, $\widetilde{W}_0 = \lambda I$, estimator $\Theta_0 = 0$.

Initialize replay buffer $\mathcal{D} = \emptyset$, neural approximator $\hat{\phi}$.

**for** $t = 1$ **to** $T$ **do**

  1. Context Generation & Prediction:

  Receive pre-serving contexts $\mathcal{X}_t = \{x_{t,k}\}_{k=1}^K$.

  Form estimated features: $\widehat{z}_{t,k} \leftarrow (x_{t,k}, \widehat{y}_{t,k})$ where $\widehat{y}_{t,k} \leftarrow \hat{\phi}_{t-1}(x_{t,k})$ for all $k \in [K]$.

  2. Arm Selection & Execution:

  Select pair $(a_t, b_t)$ based on $\widehat{z}_{t,k}$ using Equations (5) and (6).

  Plays pair $(a_t, b_t)$ and observe post-serving contexts $y_{t,a_t}, y_{t,b_t}$.

  Add $(x_{t,a_t}, y_{t,a_t})$ and $(x_{t,b_t}, y_{t,b_t})$ to $\mathcal{D}$.

  3. Feedback Processing (Delayed Outcomes):

  Update history $\mathcal{H}_t \leftarrow \{s \in [t] : s + \tau_s \leq t\}$.

  Calculate weight $\omega_t \leftarrow \min(1, \alpha/\|\Delta z_t\|_{\widetilde{V}_{t-1}^{-1}})$.

  Update $\widetilde{V}_t \leftarrow \widetilde{V}_{t-1} + \kappa \omega_t \Delta z_t \Delta z_t^\top$.

  **for** $s \in \mathcal{H}_t \setminus \mathcal{H}_{t-1}$ (newly arrived outcomes) **do**

    Observe outcome $o_s$.

    Update $\widetilde{W}_s \leftarrow \widetilde{W}_{s-1} + \kappa \omega_s \Delta z_s \Delta z_s^\top$.

  **end for**

  4. Model Updates:

  Update $\hat{\phi}_t$ by training on $\mathcal{D}$.

  Update $\Theta_t$ by minimizing weighted loss $\mathcal{L}_t$ using available outcomes.

**end for**

---

$\|\Delta z_s\|_{\widetilde{M}_t^{-1}} \leq C(\tau_t)\|\Delta z_s\|_{\widetilde{V}_t^{-1}}^2$, where $C(\tau_t)$ scales with the cumulative delay. This interaction causes the corruption-induced bias to become multiplicatively coupled with the delay factor, yielding an unfavorable $\mathcal{O}(\mathcal{C}\mathcal{D})$ term on the right-hand side of the regret bound.

Accordingly, we exploit a critical information asymmetry: while outcomes are subject to stochastic latency, the contexts (covariates) are revealed instantaneously upon action selection. Leveraging this, we propose anchoring our weighting mechanism to the full information geometry $\widetilde{V}_t$—constructed from the entire chronological sequence of contexts—rather than the partial, delay-dependent geometry $\widetilde{W}_t$. This methodological shift enables a strictly tighter regret decomposition through the following mechanisms:

1. Statistical Leverage Stability: By evaluating the statistical leverage scores of each sample relative to the global information manifold $\widetilde{V}_{s-1}$, we ensure that the importance weights defined in Equation (4) are determined a priori—at the moment of action selection. This design renders the estimation process invariant to adversarial

manipulations of outcome arrival times, as the weighting mechanism is entirely decoupled from the delay sequence $\{\tau_t\}_{t=1}^T$.

2. Structural Norm Decoupling: The core utility of the $\widetilde{V}_{t-1}^{-1}$-norm is its ability to facilitate an additive error structure, allowing corruption and delay-induced errors to be bounded independently. This is achieved by partitioning the historical samples $\{1, \ldots, t-1\}$ into three disjoint categories based on their observational status:

$$[t-1] = \underbrace{\mathcal{A}_t \cap \mathcal{E}^c}_{\text{arrived \& clean}} \sqcup \underbrace{\mathcal{A}_t \cap \mathcal{E}}_{\text{arrived \& corrupted}} \sqcup \underbrace{\mathcal{A}_t^c}_{\text{pending feedback}},$$

where $\mathcal{A}_t = \{s : s + \tau_s < t\}$ denotes the set of rounds whose feedback has arrived, and $\mathcal{E} = \{s : c_s = 1\}$ denotes the set of corrupted rounds.

By the approach above, we obtain the following concentration inequality estimation:

**Lemma 5.1** (Estimation Error Bound). *Suppose that Assumption 4.1–Assumption 4.4 are satisfied. Let $\Theta_*$ denote the true parameter and $\Theta_t$ be the estimator satisfying Equation (3). Assume the link function satisfies $\dot{g}(x) \geq \kappa > 0$. Then, for any $\delta \in (0,1)$, the following inequality holds with probability at least $1 - \delta$:*

$$\|\Theta_{t-1} - \Theta_*\|_{\widetilde{W}_{t-1}} \leq \|\Theta_{t-1} - \Theta_*\|_{\widetilde{V}_{t-1}} \leq \beta_t, \quad (7)$$

*where the confidence radius $\beta_t$ is defined as*

$$\beta_t = \frac{1}{2}\sqrt{d \log\left(\frac{1 + t/(d\lambda)}{\delta}\right)} + \sqrt{\lambda}M$$
$$+ \alpha\mathcal{C} + \alpha\mathcal{D}$$

*with $\mathcal{D} = \max(\Lambda^{1/2}, \mu_\tau)$.*

The lemma above says that

$$(\text{Estimation Error}) \leq \underbrace{\mathcal{O}(\sqrt{d})}_{\text{statistical}} + \underbrace{\mathcal{O}(\mathcal{C}\alpha)}_{\text{corruption}} + \underbrace{\mathcal{O}(\mathcal{D}\alpha)}_{\text{delay}}.$$

Using Lemma 5.1, we obtain the following regret upper bounds.

**Theorem 5.2** (Regret Upper Bound). *Suppose the conditions of Lemma 5.1 hold. By setting the exploration width $c_t = 2\beta_t$ and the parameter*

$$\alpha = \frac{\sqrt{d}}{\mathcal{C} + \mathcal{D}}, \quad (8)$$

*the cumulative regret $R_T$ satisfies*

$$R_T = \widetilde{O}\left(A_T + B_T\right),$$

*omitting sub-leading terms additionally to highlight the dominant scaling, where $A_T = d\left(\sqrt{T} + \mathcal{C} + \mathcal{D}\right)$, $B_T = T^{1-a}d_x^a(1 + \sqrt{d})$.*

*Proof Sketch.* The proof relies on a novel information-geometric decoupling that effectively separates estimation uncertainty from the regret decomposition. We distinguish between three feature variants: the estimated features $\Delta\widehat{z}_t$ used for selection, the observed noisy features $\Delta z_t$ used for weighting, and the *noise-free latent ground-truth* $\Delta z_t^*$ (where $z_{t,k}^* := (x_{t,k}, \phi_*(x_{t,k}))$).

**Step 1: Robust Confidence Bounds via Bias-Variance Trade-off.** By invoking Lemma 5.1, we establish that the estimation error is bounded by $\beta_t = \widetilde{\mathcal{O}}(\sqrt{d} + (\mathcal{C} + \mathcal{D})\alpha)$. Here, the term $(\mathcal{C} + \mathcal{D})\alpha$ quantifies the bias induced by adversarial corruption and invisible delayed feedback. We select the clipping threshold $\alpha = \sqrt{d}/(\mathcal{C} + \mathcal{D})$ to balance this bias against the variance, thereby yielding a dimension-dependent confidence radius of $\beta_t = \widetilde{\mathcal{O}}(\sqrt{d})$.

**Step 2: Regret Decomposition and Approximation Control.** Under the principle of optimism with exploration width $c_t = 2\beta_t$, the instantaneous regret is dominated by the uncertainty of the *selected* estimator: $r_t \lesssim \beta_t \|\Delta\widehat{z}_t\|_{\widetilde{V}_{t-1}^{-1}}$. By applying the triangle inequality and the learnability assumption (Assumption 4.3), we decompose this into the latent truth and the approximation error:

$$\|\Delta\widehat{z}_t\|_{\widetilde{V}_{t-1}^{-1}} \leq \|\Delta z_t^*\|_{\widetilde{V}_{t-1}^{-1}} + \|\Delta\widehat{z}_t - \Delta z_t^*\|_{\widetilde{V}_{t-1}^{-1}}.$$

The approximation error term accumulates to a sublinear order of $\widetilde{\mathcal{O}}(T^{1-a})$, reducing the problem to bounding the cumulative elliptic potential of the latent features $\sum_{t=1}^{T} \|\Delta z_t^*\|_{\widetilde{V}_{t-1}^{-1}}$.

**Step 3: Weighted Analysis.** Partitioning based on weights $\omega_t$ (derived from observed $\Delta z_t$), the unclipped regime sums to $\widetilde{\mathcal{O}}(\sqrt{dT})$. In the clipped regime ($\|\Delta z_t\|_{\widetilde{V}_{t-1}^{-1}} > \alpha$), using $\|\Delta z_t^*\| \approx \|\Delta z_t\|$, the contribution scales as $\alpha^{-1}\sum\omega_t\|\Delta z_t\|_{\widetilde{V}_{t-1}^{-1}}^2 = \widetilde{\mathcal{O}}(\alpha^{-1}d)$. Substituting $\alpha$ recovers $\widetilde{\mathcal{O}}(d(\mathcal{C} + \mathcal{D}))$. $\square$

Furthermore, we establish lower bounds by establishing worst case instances, respectively.

**Theorem 5.3** (Minimax Lower Bound). *Assume the absence of post-serving contexts. For any dimension $d$, corruption budget $\mathcal{C}$, horizon $T$, adversarial delay budget $\Lambda$, and mean stochastic delay $\mu_\tau$ satisfying the conditions $T \geq \max\{\frac{4d^2}{25}, d\mathcal{C}\}$, $\Lambda \leq T^2$, and $\mu_\tau < T$, and for any algorithm $\mathcal{A}$, there exists an instance $\mathcal{I}$ such that the expected*

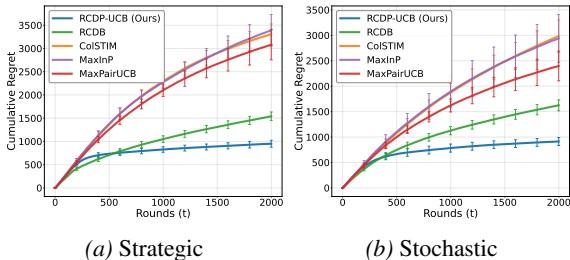

*(a)* Strategic      *(b)* Stochastic

*Figure 1.* Performance comparison without post-serving contexts, i.e., the linear setting ($\mathcal{C} = 25$, $\Lambda = 10^4$, $\mu_\tau = 100$, $\sigma = 10$; 10 runs). RCDP-UCB consistently outperforms RCDB, demonstrating the robustness of our unified weighting mechanism.

*regret is lower bounded by:*

$$\mathbb{E}[R_T(\mathcal{A}; \mathcal{I})] \geq \Omega\left(\frac{d\sqrt{T} + d\mathcal{C} + \mathcal{D}'}{\kappa}\right),$$

*omitting sub-leading terms additionally to highlight the dominant scaling, where $\mathcal{D}' := \max\{\sqrt{d\Lambda}, d\mu_\tau\}$.*

*Remark* 5.4 (Order-wise Optimality with $a = 1/2$). By setting $a = 1/2$, the regret upper bound in Theorem 5.2 can be simplified to: $R_T = \widetilde{\mathcal{O}}\left(d\left(\sqrt{T} + \mathcal{C} + \max(\Lambda^{1/2}, \mu_\tau)\right)\right)$. Neglecting the multiplicative factor of $\sqrt{d}$ for adversarial delays, this result of $R_T$ for $a = 1/2$ is remarkably consistent with the lower bounds established in Theorem 5.3. Both the upper and lower bounds exhibit a matching dependence on the primary parameters: the square-root of the time horizon $\sqrt{T}$, the linear corruption budget $C$, and the respective delay budgets $\sqrt{\Lambda}$ and $\mu_\tau$. This alignment confirms the fundamental additive costs incurred by adversarial corruption and delayed feedback.

The rigorous justifications for Lemma 5.1 (Section A), Theorem 5.2 (Section B), and Theorem 5.3 (Section C) are provided to the supplementary material.

# 6. Experimental Results

We evaluate RCDP-UCB across synthetic environments simulating the interplay of latent post-serving dynamics, feedback corruption, and observation delays.

## 6.1. Experimental Setup

We consider a contextual dueling bandit problem with $d_x = 10$ and $K = 10$ arms. Pre-serving contexts are sampled uniformly from $[-\pi, \pi]^{d_x}$. We employ three non-linear mappings $\phi_* : \mathbb{R}^{d_x} \to \mathbb{R}^{d_y}$: (i) Polynomial: $y_{t,k} \propto [x_{t,k}^2, \sqrt{|x_{t,k}|}]^\top$; (ii) Sinusoidal: $y_{t,k} \propto [\cos(x_{t,k}), \sin(x_{t,k})]^\top$; and (iii) Absolute: $y_{t,k} = |x_{t,k}|$.

The unknown mapping $\phi_*$ is approximated using a Multi-Layer Perceptron (MLP) $\hat{\phi}_t$ comprising two hidden layers

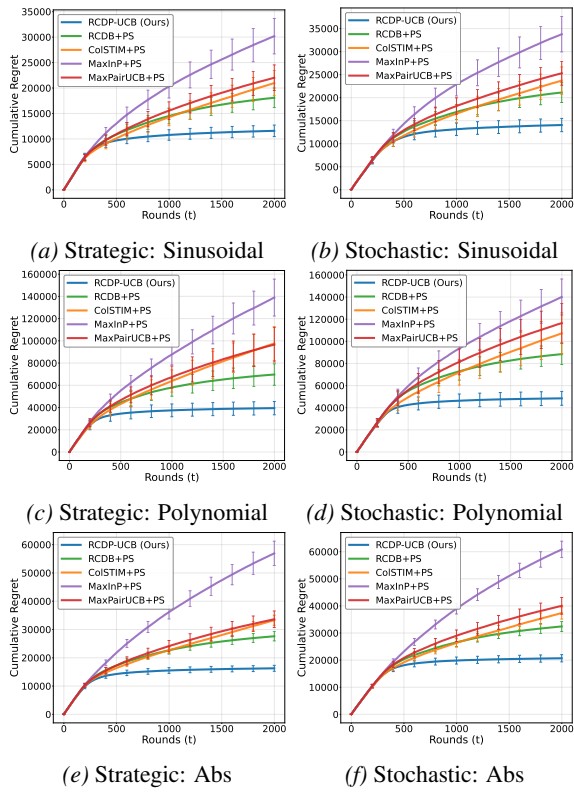

*Figure 2.* Cumulative regret with learned post-serving contexts ($\mathcal{C} = 25$, $\Lambda = 10^4$, $\mathcal{N}(100, 10^2)$). Averaged over 10 runs, RCDP-UCB consistently outperforms baselines, demonstrating superior robustness in latent environments.

(64 units each) with ReLU activations. The network is updated for 2 epochs per round using the Adam optimizer with a learning rate of $10^{-3}$, utilizing an accumulated replay buffer of observed $(x, y)$ pairs. We fix the regularization parameter $\lambda = 1.0$. We use the BTL model (Hunter, 2004; Luce, 2005) as the link function $g$. Detailed hyperparameter configurations are provided in Section D.

We investigate two delay regimes: (i) Stochastic Delay ($\tau_t \sim \mathcal{N}(\mu_\tau, \sigma^2)$) and (ii) Strategic Delay (adversarial starvation attack). We evaluate robustness under a prioritized interference protocol where the adversary favors strategic outcome corruption over delay. Subject to budget $\mathcal{C}$, corruption yields immediate falsified feedback, whereas delays affect only uncorrupted outcomes. This prioritization models a strategic adversary aiming to maximize the disruptive efficacy of the attack under the premise that direct outcome falsification impedes convergence more severely.

### 6.2. Baselines

We benchmark RCDP-UCB against state-of-the-art dueling bandit algorithms. To ensure a fair comparison, all baselines are provided with the observable pre-serving contexts $\mathcal{X}_t$. The baselines include: (i) **RCDB** (Di et al., 2025), a ro-

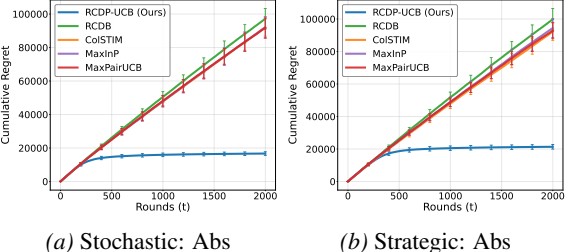

*(a) Stochastic: Abs*      *(b) Strategic: Abs*

*Figure 3.* Cumulative regret where only RCDP-UCB exploits the learned mapping $\phi_*$, while baselines are restricted to pre-serving contexts ($\mathcal{C} = 25$, $\Lambda = 10^4$, $\mathcal{N}(100, 10^2)$). Averaged over 10 runs, RCDP-UCB demonstrates superior performance relative to the baselines.

bust method utilizing weighted MLE; (ii) **ColSTIM** (Bengs et al., 2022), a randomized soft-elimination strategy; (iii) **MaxInP** (Saha, 2021), an approach based on maximizing information gain; (iv) **MaxPairUCB** (Di et al., 2023), a strategy comparing the two arms with the highest upper confidence bounds. In this section, we fix the aggregate error budget at $\mathcal{C} + \mathcal{D} = 125$, with a corruption budget of $\mathcal{C} = 25$. Accordingly, for both RCDP-UCB and RCDB, the robustness parameter $\alpha$ is pre-calibrated to be theoretically.

### 6.3. Empirical Analysis

**Efficacy of Adaptive Weighting.** Figure 1 isolates the impact of our weighting strategy by comparing RCDP-UCB with baselines in a setting without post-serving contexts (i.e., a standard linear dueling problem). RCDP-UCB generalizes the RCDB framework by incorporating a unified budget $\mathcal{C} + \mathcal{D}$ into the weight calculation (Equation (4)). RCDP-UCB consistently outperforms baselines across all delay settings. This confirms that down-weighting observations by Equation (4) effectively mitigates instability from both corruption and delay.

**Robustness in Latent Post-Serving Environments.** Figure 2 presents the cumulative regret in environments with non-linear post-serving mappings. Here, all algorithms (including baselines) are augmented to learn $\phi_*$. RCDP-UCB achieves the lowest cumulative regret across all tasks (Absolute, Polynomial, Sinusoidal) independently of delay regimes. This demonstrates that our algorithm successfully leverages predicted post-serving contexts while neutralizing the adverse effects of delayed and corrupted feedback.

**Impact of Post-Serving Context Awareness.** Figure 3 further investigates the scenario where only RCDP-UCB learns and utilizes the post-serving mapping $\phi_*$, while baselines rely solely on pre-serving contexts. As expected, RCDP-UCB typically outperforms the baselines. This highlights the critical advantage of explicitly modeling and utilizing latent post-serving dynamics, provided the learning

algorithm is robust enough to handle the associated corruptions and delays.

Further experimental results, including sensitivity analyses regarding context dimensionality, arm set size, and error budgets, alongside evaluations on real-world datasets, are detailed in Section E of the supplementary material.

## 7. Conclusion and Discussion

In this work, we introduced RCDP-UCB, a unified framework for linear contextual dueling bandits designed to robustly handle latent post-serving contexts, adversarial corruptions, and unknown feedback delays. By anchoring our analysis to the full information design matrix $\widetilde{V}_{t-1}$, we developed an adaptive weighting mechanism operating on a unified error budget $\mathcal{C} + \mathcal{D}$. This design enables the algorithm to determine importance weights *a priori*, effectively decoupling parameter estimation from the specific realization of delay sequences. Theoretically, RCDP-UCB achieves sublinear regret while remaining agnostic to the underlying delay regime; our bounds match information-theoretic lower bounds up to logarithmic factors in the stochastic setting and remain near-optimal within a $\sqrt{d}$ factor in the adversarial delay regime.

**Limitations.** A critical direction for future research is extending this framework to handle delayed post-serving contexts. Additionally, closing the $\mathcal{O}(\sqrt{d})$ gap in the adversarial delay lower bound remains an open problem. Additionally, establishing a tight information-theoretic lower bound that explicitly accounts for the learnability of post-serving mappings remains an open problem.

## Impact Statement

This paper presents a robust algorithmic framework for interactive decision-making, with significant implications for enhancing the reliability of systems reliant on human feedback, such as recommendation engines and Reinforcement Learning from Human Feedback (RLHF). By effectively mitigating the risks associated with adversarial data poisoning and feedback delays, our work contributes to the development of safer AI systems that are resilient to manipulation. Furthermore, the integration of latent post-serving contexts promotes a more accurate alignment with genuine user intent, moving beyond superficial engagement metrics. While our primary focus is on robustness, we encourage practitioners to carefully calibrate these mechanisms to ensure that legitimate minority feedback is not inadvertently filtered out as adversarial noise.

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

# Supplementary Material:

# Robust Linear Dueling Bandits with Post-serving Context under Unknown Delays and Adversarial Corruptions

In Section A, we derive the concentration bound for parameter estimation error under the dual challenges of adversarial corruptions and delayed feedback. We then provide the complete proof for the regret upper bound of Theorem 5.2 in Section B, followed by the proof of the lower bound (Theorem 5.3) in Section C. Finally, in Section D, we provide implementation details for the experimental results in Section 6, ablation studies varying the dimension of arms and contexts, the corruption budget $\mathcal{C}$, and delay complexity $\mathcal{D}$, as well as additional results regarding hyperparameter sensitivity and baseline comparisons.

## A. Proof of Lemma 5.1

The following lemma is a weighted generalization of the self-normalized bound for vector-valued martingales (Abbasi-Yadkori et al., 2011).

**Lemma A.1** (Weighted Self-Normalized Bound). *Let $\{\mathcal{F}_t\}_{t=0}^{\infty}$ be a filtration. Let $\{\varepsilon_s\}_{s\geq 1}$ be a real-valued process such that $\varepsilon_s$ is $\mathcal{F}_s$-measurable and conditionally R-sub-Gaussian. Let $\{\Delta z_s\}_{s\geq 1}$ be an $\mathbb{R}^{\bar{d}}$-valued process where $\Delta z_s$ is $\mathcal{F}_{s-1}$-measurable and $\|\Delta z_s\|_2 \leq L$. For any sequence of predictable weights $\omega_s \in (0,1]$ and scaling factor $\kappa > 0$, define $\mathcal{S}_t = \sum_{s=1}^{t} \omega_s \varepsilon_s \Delta z_s$ and $\widetilde{V}_t = \lambda I + \kappa \sum_{s=1}^{t} \omega_s \Delta z_s \Delta z_s^\top$. Then, for any $\delta > 0$, with probability at least $1 - \delta$, the following inequality holds for all $t \geq 1$:*

$$\|\mathcal{S}_t\|_{\widetilde{V}_t^{-1}} \leq \frac{R\sqrt{\bar{\omega}}}{\sqrt{\kappa}} \sqrt{2 \log\left(\frac{\det(\widetilde{V}_t)^{1/2}}{\delta \det(\lambda I)^{1/2}}\right)},$$

*where $\bar{\omega} = \max_{s \leq t} \omega_s$. By the determinant-trace inequality, it holds that:*

$$\|\mathcal{S}_t\|_{\widetilde{V}_t^{-1}} \leq \frac{R\sqrt{\bar{\omega}}}{\sqrt{\kappa}} \sqrt{d \log\left(1 + \frac{\kappa t L^2 \bar{\omega}}{d\lambda}\right) + 2 \log\left(\frac{1}{\delta}\right)}.$$

*Proof.* Define the transformed covariates $X_s := \sqrt{\kappa \omega_s} \Delta z_s$ and the scaled noise terms $\epsilon_s := \sqrt{\omega_s/\kappa}\, \varepsilon_s$. Under the assumption that $\omega_s$ is $\mathcal{F}_{s-1}$-measurable and $\omega_s \in (0,1]$, $\epsilon_s$ is conditionally $R'$-sub-Gaussian with $R' = R\sqrt{\omega_s/\kappa} \leq R\sqrt{\bar{\omega}/\kappa}$. The martingale sum and the design matrix then take the standard forms:

$$\mathcal{S}_t = \sum_{s=1}^{t} \epsilon_s X_s, \quad \text{and} \quad \widetilde{V}_t = \lambda I + \sum_{s=1}^{t} X_s X_s^\top.$$

Directly applying Theorem 1 from Abbasi-Yadkori et al. (2011) to the pair $(X_s, \epsilon_s)$ with the uniform sub-Gaussian proxy $R\sqrt{\bar{\omega}/\kappa}$ yields the self-normalized bound. The final result follows by substituting the determinant-trace inequality $\log(\det(\widetilde{V}_t)/\det(\lambda I)) \leq d \log(1 + \frac{\kappa t L^2 \bar{\omega}}{d\lambda})$. $\qquad\square$

**Lemma A.2** (Bound on Invisible Feedbacks). *Let $D_t := \sum_{s=1}^{t-1} \mathbb{I}_{s+\tau_s > t}$ denote the number of invisible feedbacks at time $t$, where $\tau_s$ is the delay associated with the feedback from round s. Let $\Lambda$ denote the cumulative delay budget, and let $\mu_\tau$ denote the mean of sub-Gaussian delays. Then, for any $t \in [T]$, with probability at least $1 - \delta$:*

$$D_t = \widetilde{\mathcal{O}}(\mathcal{D}), \quad \text{where} \quad \mathcal{D} := \max\left(\sqrt{\Lambda}, \max(\mu_\tau, 1)\right). \tag{A.1}$$

*Proof.* We consider the two delay models separately.

**Stochastic Delays.** By Lemma 4 in (Howson et al., 2023), for sub-Gaussian delays:

$$D_t = \widetilde{\mathcal{O}}\left(\mu_\tau + \sqrt{\mu_\tau \log(1/\delta)} + 1\right) = \widetilde{\mathcal{O}}(\max(\mu_\tau, 1)). \tag{A.2}$$

**Adversarial Delays.** Let $\mathcal{S}_t := \{s < t : s + \tau_s > t\}$ denote the set of indices with pending feedback at time $t$, so that $D_t = |\mathcal{S}_t|$. For each $s \in \mathcal{S}_t$, the feedback is invisible only if the delay satisfies $\tau_s > t - s$. Since delays are integer-valued, this implies $\tau_s \geq t - s + 1$.

The total delay budget provides a lower bound:

$$\Lambda = \sum_{s=1}^{T} \tau_s \geq \sum_{s \in \mathcal{S}_t} \tau_s \geq \sum_{s \in \mathcal{S}_t} (t - s + 1).$$

To minimize this sum for a fixed cardinality $D_t$, the adversary acts greedily by choosing the indices closest to $t$, namely $s \in \{t - 1, t - 2, \ldots, t - D_t\}$. Substituting $k = t - s$, the sum becomes:

$$\Lambda \geq \sum_{k=1}^{D_t} (k + 1) = \frac{D_t(D_t + 1)}{2} + D_t = \frac{D_t^2 + 3D_t}{2} \geq \frac{D_t^2}{2}.$$

Inverting this quadratic inequality yields the bound on the number of invisible feedbacks:

$$D_t \leq \sqrt{2\Lambda}. \tag{A.3}$$

Combining Eq. (A.2) and Eq. (A.3) yields Eq. (A.1). $\qquad\square$

We define by introducing auxiliary quantities that will be instrumental in our analysis:

$$\eta_t := l_t - g(\Delta z_t^\top \Theta_*),$$
$$\zeta_t := \gamma_t - g(\Delta z_t^\top \Theta_*),$$

where $l_t$ denotes the unobserved true feedback, $\gamma_t$ represents the corrupted observation, $g(\cdot)$ is the link function, and $\Theta_*$ is the true underlying parameter. Additionally, we define the auxiliary function $G_t : \mathbb{R}^d \to \mathbb{R}^d$ as:

$$G_t(\Theta) := \lambda\Theta + \sum_{s=1}^{t-1} \omega_s \big(g(\Theta^\top \Delta z_s) - g(\Theta_*^\top \Delta z_s)\big)\Delta z_s,$$

where $\omega_s$ denotes the adaptive weight assigned to the $s$-th observation as Equation (4).

**Step 1: Error Decomposition via Local Strong Convexity**

By the Mean Value Theorem, for any $\Theta_1, \Theta_2 \in \mathbb{R}^d$, there exists $\overline{\Theta}$ on the line segment connecting them such that:

$$G_t(\Theta_1) - G_t(\Theta_2) = \nabla G_t(\overline{\Theta})(\Theta_1 - \Theta_2) = H_t(\overline{\Theta})(\Theta_1 - \Theta_2),$$

where the Hessian matrix $H_t(\overline{\Theta})$ is given by:

$$H_t(\overline{\Theta}) := \lambda I + \sum_{s=1}^{t-1} \omega_s \dot{g}(\overline{\Theta}^\top \Delta z_s)\Delta z_s \Delta z_s^\top.$$

with $\lambda > 0$. Since $\dot{g}(\cdot) \geq \kappa > 0$ by Assumption 4.1, we have the matrix domination $H_t(\overline{\Theta}) \succeq \widetilde{V}_{t-1}$, where:

$$\widetilde{V}_{t-1} := \lambda I + \kappa \sum_{s=1}^{t-1} \omega_s \Delta z_s \Delta z_s^\top.$$

as in Equation (1). Utilizing the identity $\|Ax\|_{A^{-1}} = \|x\|_A$ for any positive definite matrix $A$, we obtain the following monotonicity property:

$$\|G_t(\Theta_1) - G_t(\Theta_2)\|_{\widetilde{V}_{t-1}^{-1}} \geq \|\Theta_1 - \Theta_2\|_{\widetilde{V}_{t-1}}. \tag{A.4}$$

Note that

$$G_t(\Theta_*) = \lambda\Theta_*$$

by the definition of $G_t$. Let $\Theta_t$ be the estimator satisfying Eq. (3). By the optimality condition, we have:

$$\lambda\Theta_t + \sum_{s=1}^{t-1} \omega_s \mathbb{I}_{s+\tau_s \leq t}\big(g(\Theta_t^\top \Delta z_s) - o_s\big)\Delta z_s = 0, \tag{A.5}$$

where $\mathbb{I}_{s+\tau_s \leq t}$ is the indicator function for whether the feedback from round $s$ has arrived by time $t$, and $o_s$ denotes the observed (potentially corrupted) feedback.

Using Eq. (A.5), we decompose $G_t(\Theta_t)$ into three components:

$$\begin{aligned} G_t(\Theta_t) &= \lambda\Theta_t + \sum_{s=1}^{t-1} \omega_s\big(g(\Theta_t^\top \Delta z_s) - g(\Theta_*^\top \Delta z_s)\big)\Delta z_s \\ &= \underbrace{\sum_{s=1}^{t-1} \mathbb{I}_{s+\tau_s \leq t}\omega_s\big(o_s - g(\Theta_*^\top \Delta z_s)\big)\Delta z_s}_{S_{t,1}} \\ &\quad + \underbrace{\sum_{s=1}^{t-1}(1 - \mathbb{I}_{s+\tau_s \leq t})\omega_s\big(g(\Theta_t^\top \Delta z_s) - o_s\big)\Delta z_s}_{S_{t,2}} \\ &\quad - \underbrace{\sum_{s=1}^{t-1}(1 - \mathbb{I}_{s+\tau_s \leq t})\omega_s\big(g(\Theta_*^\top \Delta z_s) - o_s\big)\Delta z_s}_{S_{t,3}}. \end{aligned} \tag{A.6}$$

In Equation (A.6), the term $S_{t,1}$ captures the contribution from observed feedback, while $S_{t,2}$ and $S_{t,3}$ account for the missing information due to delayed feedback.

**Step 2: Calculation of $S_{t,1}$**

Let $C_s \in \{0,1\}$ denote the corruption indicator at round $s$, where $C_s = 1$ indicates that the feedback has been corrupted (unbeknownst to the learner). We decompose $S_{t,1}$ as:

$$\begin{aligned} S_{t,1} &= \sum_{\substack{s=1 \\ C_s=0}}^{t-1} \mathbb{I}_{s+\tau_s \leq t}\omega_s\eta_s\Delta z_s + \sum_{\substack{s=1 \\ C_s=1}}^{t-1} \mathbb{I}_{s+\tau_s \leq t}\omega_s(\gamma_s - \eta_s + \eta_s)\Delta z_s \\ &= \sum_{s=1}^{t-1} \mathbb{I}_{s+\tau_s \leq t}\omega_s\eta_s\Delta z_s + \sum_{\substack{s=1 \\ C_s=1}}^{t-1} \mathbb{I}_{s+\tau_s \leq t}\omega_s(\gamma_s - \eta_s)\Delta z_s \\ &=: S_{t,1,1} + S_{t,1,2}. \end{aligned}$$

The term $S_{t,1,1}$ corresponds to the stochastic noise contribution, while $S_{t,1,2}$ captures the adversarial corruption effect.

A key technical challenge arises because $S_{t,1,1}$ is not a martingale when delays are chosen adaptively by an adversary—the terms $\mathbb{E}[\eta_s]$ and $\mathbb{I}_{s+\tau_s \leq t}$ may exhibit dependence. To address this, we perform a further decomposition:

$$S_{t,1,1} = \underbrace{\sum_{s=1}^{t-1} \omega_s\eta_s\Delta z_s}_{S_{t,1,1,1}} - \underbrace{\sum_{s=1}^{t-1} \mathbb{I}_{s+\tau_s > t}\omega_s\eta_s\Delta z_s}_{S_{t,1,1,2}}.$$

The term $S_{t,1,1,1}$ is now a proper martingale with respect to the natural filtration, while $S_{t,1,1,2}$ captures the contribution from observations whose feedback has not yet arrived.

Applying Lemma A.1 to $S_{t,1,1,1}$ with $R = 1/2$ (corresponding to the Gaussian prior $\mathcal{N}(0, \lambda^{-1}I)$ settings) and $\bar{\omega} = 1$, we obtain that with probability at least $1 - \delta$:

$$\|S_{t,1,1,1}\|_{\widetilde{V}_t^{-1}} \leq C\sqrt{d \log\left(\frac{1 + tL^2/(d\lambda)}{\delta}\right)}.$$

for some constant $C = C(\kappa) > 0$.

For the delayed feedback term $S_{t,1,1,2}$, note that $\eta_s$ may not be observed to the leaner yet. However, we already know that $|\eta_s| \leq 1$. Accordingly, we have:

$$\|S_{t,1,1,2}\|_{\widetilde{V}_t^{-1}} \leq \sum_{s=1}^{t-1} \mathbb{I}_{s+\tau_s > t} \omega_s \|\Delta z_s\|_{\widetilde{V}_t^{-1}} \leq \alpha D_t,$$

due to $\omega_s = \min\left(1, \alpha/\|\Delta z_s\|_{\widetilde{V}_{s-1}^{-1}}\right)$, where $D_t := \sum_{s=1}^{t-1} \mathbb{I}_{s+\tau_s > t}$ denotes the number of invisible (pending) feedbacks at time $t$. Thus,

$$\|S_{t,1,1}\|_{\widetilde{V}_t^{-1}} \leq C\sqrt{d \log\left(\frac{1 + tL^2/(d\lambda)}{\delta}\right)} + \alpha D_t. \tag{A.7}$$

For the corruption term $S_{t,1,2}$, the design of adaptive weights $\omega_s = \min\left(1, \alpha/\|\Delta z_s\|_{\widetilde{V}_{s-1}^{-1}}\right)$ yields:

$$\|S_{t,1,2}\|_{\widetilde{V}_{t-1}^{-1}} \leq \sum_{\substack{s=1 \\ C_s=1}}^{t-1} \omega_s \|\Delta z_s\|_{\widetilde{V}_{t-1}^{-1}} \leq \mathcal{C}\alpha, \tag{A.8}$$

where $\mathcal{C}$ denotes the total corruption budget. As a result, combining Equations (A.7) and (A.8), we obtain

$$\|S_{t,1}\|_{\widetilde{V}_{t-1}^{-1}} \leq C\sqrt{d \log\left(\frac{1 + tL^2/(d\lambda)}{\delta}\right)} + \alpha\mathcal{C} + \alpha D_t. \tag{A.9}$$

**Step 2: Calculation of $S_{t,2}$ and $S_{t,3}$**

For the terms $S_{t,2}$ and $S_{t,3}$ arising from pending feedback, using $\omega_s > 0$ and $|g(\Theta_*^\top \Delta z_s) - o_s| \leq 1$, we obtain:

$$\|S_{t,2}\|_{\widetilde{V}_{t-1}^{-1}} + \|S_{t,3}\|_{\widetilde{V}_{t-1}^{-1}} \leq 2\sum_{s=1}^{t-1} \mathbb{I}_{s+\tau_s > t} \omega_s \|\Delta z_s\|_{\widetilde{V}_{t-1}^{-1}}$$

$$\leq 2\alpha \sum_{s=1}^{t-1} \mathbb{I}_{s+\tau_s > t} = 2\alpha D_t. \tag{A.10}$$

**Step 3: Final Error Bound**

Utilizing Lemma A.2, we can obtain

$$\alpha D_t = \alpha \widetilde{O}(\mathcal{D}).$$

where $\mathcal{D} = \max(\Lambda^{1/2}, \mu_\tau)$. Accordingly, setting the robustness parameter $\alpha$ as:

$$\alpha = \frac{\sqrt{d}}{\mathcal{C} + \mathcal{D}},$$

and combining the bounds from Eqs. (A.4), (A.9) and (A.10), we obtain:

$$\|\Theta_{t-1} - \Theta_*\|_{\widetilde{V}_{t-1}} = \widetilde{\mathcal{O}}(\sqrt{d}).$$

due to Lemma A.2. Finally, since $\widetilde{W}_{t-1} \preceq \widetilde{V}_{t-1}$, we have:

$$\|\Theta_{t-1} - \Theta_*\|_{\widetilde{W}_{t-1}} \leq \|\Theta_{t-1} - \Theta_*\|_{\widetilde{V}_{t-1}} \leq \beta_t,$$

where the confidence radius is given by:

$$\beta_t := \frac{1}{2}\sqrt{d \log\left(\frac{1 + tL^2/(d\lambda)}{\delta}\right)} + \sqrt{d}\frac{\mathcal{C}}{\mathcal{C} + \mathcal{D}} + \sqrt{\lambda}\|\Theta_*\| + \sqrt{d}\frac{3\mathcal{D}}{\mathcal{C} + \mathcal{D}}.$$

The form $\|\Theta_* - \Theta_{t-1}\|_{\widetilde{V}_{t-1}} \leq \beta_t$ is exactly the statement used in Section B and aligns with the arm-selection rule employing $\Theta_{t-1}$ at round $t$.

This completes the proof. $\qquad\qquad\qquad\qquad\qquad\qquad\qquad\qquad\qquad\qquad\qquad\qquad\qquad\qquad\qquad\square$

## B. Proof of Theorem 5.2

We first state the generalized elliptic potential lemma.

**Lemma B.1** (Generalized Elliptical Potential Lemma (Wang et al., 2024))**.** *Suppose*

1. *$X_0 \in \mathbb{R}^{d \times d}$ is any positive definite matrix;*

2. *$x_1, \ldots, x_T \in \mathbb{R}^d$ is a sequence of vectors with bounded $\ell_2$ norm $\max_t \|x_t\| \leq L_x$;*

3. *$\epsilon_1, \ldots, \epsilon_T \in \mathbb{R}^d$ is a sequence of independent (not necessarily identical) bounded zero-mean noises satisfying $\max_t \|\epsilon_t\| \leq L_\epsilon$ and $\mathbb{E}[\epsilon_t \epsilon_t^\top] \succeq \sigma_\epsilon^2 I$ for any $t$; and*

4. *$\widetilde{X}_t$ is defined as follows:*
$$\widetilde{X}_t = X_0 + \sum_{s=1}^{t}(x_s + \epsilon_s)(x_s + \epsilon_s)^\top \in \mathbb{R}^{d \times d}.$$

*Then, for any $p \in [0, 1]$, the following inequality holds with probability at least $1 - \delta$,*

$$\sum_{t=1}^{T}\left(1 \wedge \|x_t\|_{\widetilde{X}_{t-1}^{-1}}^2\right)^p \leq 2^p T^{1-p} \log^p\left(\frac{\det X_T}{\det X_0}\right) + \frac{8L_\epsilon^2(L_\epsilon + L_x)^2}{\sigma_\epsilon^4}\log\left(\frac{32dL_\epsilon^2(L_\epsilon + L_x)^2}{\delta\sigma_\epsilon^4}\right)$$

*where $a \wedge b = \min\{a, b\}$.*

For clarity, we define the key quantities used throughout the proof. For each round $t \in [T]$ and arm $k \in [K]$, we define three variants of the joint feature vector:

$$\begin{aligned}
\widehat{z}_{t,k} &:= \left(x_{t,k}, \widehat{\phi}_t(x_{t,k})\right), \\
z_{t,k}^* &:= \left(x_{t,k}, \phi_*(x_{t,k})\right), \\
z_{t,k} &:= \left(x_{t,k}, y_{t,k}\right),
\end{aligned}$$

where $\widehat{\phi}_t$ denotes the learner's estimate of the true post-serving function $\phi_*$ at round $t$, and $y_{t,k}$ is the observed post-serving context. Additionally, we use the shorthand notation $\Delta z_{t,k,k'} := z_{t,k} - z_{t,k'}$ for differences between feature vectors, and analogously for $\Delta\widehat{z}_{t,k,k'}$ and $\Delta z_{t,k,k'}^*$.

### Step 1: Analysis of Instantaneous Regret

The proof proceeds by decomposing the instantaneous regret into three interpretable components and bounding each term separately. Let $k_t^*$ denote the optimal arm at round $t$, and let $a_t$ and $b_t$ denote the arms selected by our algorithm according to Equation (5)–Equation (6). The instantaneous regret $r_t$ satisfies:

$$2r_t = \langle\Theta_*, \Delta z_{t,k_t^*,a_t}^* + \Delta z_{t,k_t^*,b_t}^*\rangle.$$

We decompose this quantity into three components by adding and subtracting appropriate terms:

$$2r_t = \underbrace{\langle \Theta_{t-1}, \Delta \widehat{z}_{t,k_t^*,a_t} + \Delta \widehat{z}_{t,k_t^*,b_t} \rangle}_{\mathcal{B}_{t,1}:\text{Confidence Interval Term}}$$

$$+ \underbrace{\langle \Theta_*, (\Delta z_{t,k_t^*,a_t}^* - \Delta \widehat{z}_{t,k_t^*,a_t}) + (\Delta z_{t,k_t^*,b_t}^* - \Delta \widehat{z}_{t,k_t^*,b_t}) \rangle}_{\mathcal{B}_{t,2}:\text{Approximation Error Term}}$$

$$+ \underbrace{\langle \Theta_* - \Theta_{t-1}, \Delta \widehat{z}_{t,k_t^*,a_t} + \Delta \widehat{z}_{t,k_t^*,b_t} \rangle}_{\mathcal{B}_{t,3}:\text{Estimation Error Term}}.$$

The first term $\mathcal{B}_{t,1}$ captures the suboptimality arising from the algorithm's arm selection rule under the current parameter estimate. The second term $\mathcal{B}_{t,2}$ quantifies the regret induced by the discrepancy between the true post-serving function and its approximation. The third term $\mathcal{B}_{t,3}$ reflects the estimation error in the parameter.

### Step 1: Bounding the Confidence Interval Term $\mathcal{B}_{t,1}$

To estimate $\mathcal{B}_{t,1}$, we consider the arm selection strategy specified in Equation (5)–Equation (6). The selected arms $a_t$ and $b_t$ satisfy:

$$\langle \Theta_{t-1}, \widehat{z}_{t,k_t^*} \rangle \leq \langle \Theta_{t-1}, \widehat{z}_{t,a_t} \rangle, \tag{B.11}$$

$$\langle \Theta_{t-1}, \widehat{z}_{t,k} \rangle + c_t \|\Delta \widehat{z}_{t,k,a_t}\|_{\widetilde{V}_{t-1}^{-1}} \leq \langle \Theta_{t-1}, \widehat{z}_{t,b_t} \rangle + c_t \|\Delta \widehat{z}_{t,b_t,a_t}\|_{\widetilde{V}_{t-1}^{-1}}, \tag{B.12}$$

for all $k \in \mathcal{K}_t$. Using the identity

$$\Delta \widehat{z}_{t,k_t^*,a_t} + \Delta \widehat{z}_{t,k_t^*,b_t} = 2\Delta \widehat{z}_{t,k_t^*,a_t} + \Delta \widehat{z}_{t,a_t,b_t} \tag{B.13}$$

and applying (B.11)–(B.12), we obtain:

$$\begin{aligned} \mathcal{B}_{t,1} &= \langle \Theta_{t-1}, 2\Delta \widehat{z}_{t,k_t^*,a_t} + \Delta \widehat{z}_{t,a_t,b_t} \rangle \\ &\leq \langle \Theta_{t-1}, \Delta \widehat{z}_{t,k_t^*,a_t} \rangle + \langle \Theta_{t-1}, \Delta \widehat{z}_{t,a_t,b_t} \rangle \\ &\leq c_t \big( \|\Delta \widehat{z}_{t,b_t,a_t}\|_{\widetilde{V}_{t-1}^{-1}} - \|\Delta \widehat{z}_{t,k_t^*,a_t}\|_{\widetilde{V}_{t-1}^{-1}} \big). \end{aligned} \tag{B.14}$$

The term $\mathcal{B}_{t,2}$ captures the regret induced by the discrepancy between the true post-serving contexts $z_{t,k}^*$ and the learner's approximations $\widehat{z}_{t,k}$. Observe that by construction:

$$z_{t,k}^* - \widehat{z}_{t,k} = \big( 0, \phi_*(x_{t,k}) - \widehat{\phi}_t(x_{t,k}) \big).$$

Applying the Cauchy–Schwarz inequality and Assumption 4.2, we have for any arm $k$:

$$\big| \langle \Theta_*, z_{t,k}^* - \widehat{z}_{t,k} \rangle \big| \leq \|\Theta_*\|_2 \cdot \|\phi_*(x_{t,k}) - \widehat{\phi}_t(x_{t,k})\|_2 \leq M \|\phi_*(x_{t,k}) - \widehat{\phi}_t(x_{t,k})\|_2.$$

By Assumption 4.3 (General Learnability), with probability at least $1 - \delta$, the approximation error satisfies:

$$\|\phi_*(x_{t,k}) - \widehat{\phi}_t(x_{t,k})\|_2 \leq C_0 \sqrt{d_x} \, t^{-a} \log(t/\delta),$$

for some universal constant $C_0 > 0$. Combining these bounds and applying the triangle inequality over both arm pairs, we obtain:

$$\mathcal{B}_{t,2} \leq 4MC_0 \sqrt{d_x} \, t^{-a} \log(t/\delta). \tag{B.15}$$

For the estimation error term $\mathcal{B}_{t,3}$, we apply the confidence ellipsoid property. By Lemma 5.1, the parameter estimate $\Theta_{t-1}$ used by the algorithm at round $t$ (see Equation (5)–Equation (6)) satisfies:

$$\|\Theta_* - \Theta_{t-1}\|_{\widetilde{V}_{t-1}} \leq \beta_t,$$

where $\beta_t$ is the confidence radius. Crucially, this bound shows that $\|\Theta_* - \Theta_{t-1}\|_{\widetilde{V}_{t-1}}$ is controlled by $\beta_t$ uniformly in $t$, which directly bounds the first factor in the Cauchy–Schwarz decomposition of $\mathcal{B}_{t,3}$ below. Applying this bound together with the Cauchy–Schwarz inequality and Equation (B.13) yields:

$$
\begin{aligned}
\mathcal{B}_{t,3} &= \langle \Theta_* - \Theta_{t-1}, \Delta\widehat{z}_{t,k_t^*,a_t} + \Delta\widehat{z}_{t,k_t^*,b_t}\rangle \\
&\leq \|\Theta_* - \Theta_{t-1}\|_{\widetilde{V}_{t-1}}\big(2\|\Delta\widehat{z}_{t,k_t^*,a_t}\|_{\widetilde{V}_{t-1}^{-1}} + \|\Delta\widehat{z}_{t,a_t,b_t}\|_{\widetilde{V}_{t-1}^{-1}}\big) \\
&\leq 2\beta_t\|\Delta\widehat{z}_{t,k_t^*,a_t}\|_{\widetilde{V}_{t-1}^{-1}} + \beta_t\|\Delta\widehat{z}_{t,a_t,b_t}\|_{\widetilde{V}_{t-1}^{-1}}.
\end{aligned}
\tag{B.16}
$$

Setting $c_t = 2\beta_t$ and combining (B.14), (B.15), and (B.16), the instantaneous regret satisfies:

$$
2r_t \leq 3\beta_t\|\Delta\widehat{z}_{t,a_t,b_t}\|_{\widetilde{V}_{t-1}^{-1}} + 4C_0 M\sqrt{d_x}\,t^{-a}\log(t/\delta).
$$

**Step 2: Cumulative Regret Analysis**

Summing over all rounds $t \in [T]$, the cumulative regret $R_T = \sum_{t=1}^{T} r_t$ satisfies:

$$
2R_T \leq 3\beta_T \sum_{t=1}^{T}\|\Delta\widehat{z}_{t,a_t,b_t}\|_{\widetilde{V}_{t-1}^{-1}} + 4C_0 M\sqrt{d_x}\log(T/\delta)\sum_{t=1}^{T} t^{-a}.
\tag{B.17}
$$

We further decompose the first sum by relating the approximated features to the true features:

$$
\sum_{t=1}^{T}\|\Delta\widehat{z}_{t,a_t,b_t}\|_{\widetilde{V}_{t-1}^{-1}} \leq \sum_{t=1}^{T}\|\Delta z_{t,a_t,b_t}^*\|_{\widetilde{V}_{t-1}^{-1}} + \sum_{t=1}^{T}\|\Delta z_{t,a_t,b_t}^* - \Delta\widehat{z}_{t,a_t,b_t}\|_{\widetilde{V}_{t-1}^{-1}}.
$$

By the eigenvalue bound $\|\cdot\|_{\widetilde{V}_{t-1}^{-1}} \leq \lambda^{-1/2}\|\cdot\|_2$ and Assumption 4.3:

$$
\begin{aligned}
\sum_{t=1}^{T}\|\Delta z_{t,a_t,b_t}^* - \Delta\widehat{z}_{t,a_t,b_t}\|_{\widetilde{V}_{t-1}^{-1}} &\leq \frac{1}{\sqrt{\lambda}}\sum_{t=1}^{T}\|\Delta z_{t,a_t,b_t}^* - \Delta\widehat{z}_{t,a_t,b_t}\|_2 \\
&\leq \frac{2}{\sqrt{\lambda}}\sum_{t=1}^{T}\big(\|\widehat{\phi}_t(x_{t,a_t}) - \phi_*(x_{t,a_t})\|_2 + \|\widehat{\phi}_t(x_{t,b_t}) - \phi_*(x_{t,b_t})\|_2\big) \\
&\leq \frac{4C_0\sqrt{d_x}}{\sqrt{\lambda}}\log(T/\delta)\sum_{t=1}^{T} t^{-a}.
\end{aligned}
$$

For $\alpha \in (0,1)$, we employ the integral comparison:

$$
\sum_{t=1}^{T} t^{-a} \leq \int_0^T x^{-\alpha}\,dx = \frac{T^{1-\alpha}}{1-\alpha}.
$$

Thus,

$$
\sum_{t=1}^{T}\|\Delta z_{t,a_t,b_t}^* - \Delta\widehat{z}_{t,a_t,b_t}\|_{\widetilde{V}_{t-1}^{-1}} \leq \frac{4C_0\sqrt{d_x}}{\sqrt{\lambda}}\log(T/\delta)\frac{T^{a-\alpha}}{1-\alpha}.
$$

We partition the sum based on the weighting scheme:

$$
\begin{aligned}
\sum_{t=1}^{T} \|\Delta z_{t,a_t,b_t}^*\|_{\widetilde{V}_{t-1}^{-1}} &= \sum_{t:\omega_t=1} \|\Delta z_{t,a_t,b_t}^*\|_{\widetilde{V}_{t-1}^{-1}} + \sum_{t:\omega_t<1} \|\Delta z_{t,a_t,b_t}^*\|_{\widetilde{V}_{t-1}^{-1}} \\
&\leq \sum_{t:\omega_t=1} \|\Delta z_{t,a_t,b_t}^*\|_{\widetilde{V}_{t-1}^{-1}} + \frac{1}{\alpha} \sum_{t:\omega_t<1} \omega_t \|\Delta z_{t,a_t,b_t}^*\|_{\widetilde{V}_{t-1}^{-1}} \|\Delta z_{t,a_t,b_t}\|_{\widetilde{V}_{t-1}^{-1}} \\
&\leq \sum_{t:\omega_t=1} \|\Delta z_{t,a_t,b_t}^*\|_{\widetilde{V}_{t-1}^{-1}} + \frac{1}{\alpha} \left( \sum_{t:\omega_t<1} \omega_t \|\Delta z_{t,a_t,b_t}^*\|_{\widetilde{V}_{t-1}^{-1}}^2 + \sum_{t:\omega_t<1} \omega_t \|\Delta z_{t,a_t,b_t}\|_{\widetilde{V}_{t-1}^{-1}}^2 \right) \\
&\leq \sum_{t:\omega_t=1} \|\Delta z_{t,a_t,b_t}^*\|_{\widetilde{V}_{t-1}^{-1}} + \frac{\mathcal{C}+\mathcal{D}}{\sqrt{d}} \sum_{t:\omega_t<1} \|\omega_t^{1/2}\Delta z_{t,a_t,b_t}^*\|_{\widetilde{V}_{t-1}^{-1}}^2 \\
&\quad + \frac{\mathcal{C}+\mathcal{D}}{\sqrt{d}} \sum_{t:\omega_t<1} \|\omega_t^{1/2}\Delta z_{t,a_t,b_t}\|_{\widetilde{V}_{t-1}^{-1}}^2 .
\end{aligned}
\tag{B.18}
$$

$\sum_{t:\omega_t<1} \|\omega_t^{1/2}\Delta z_{t,a_t,b_t}\|_{\widetilde{V}_{t-1}^{-1}}^2$ is bounded as $\widetilde{O}(d)$ by a standard elliptic potential lemma (Lattimore & Szepesvári, 2020). In the case of $\sum_{t:\omega_t<1} \|\omega_t^{1/2}\Delta z_{t,a_t,b_t}^*\|_{\widetilde{V}_{t-1}^{-1}}^2$, we will use the generalized elliptic potential lemma:

By Lemma B.1, the upper bound of the potential sum depends on the choice of $p \in [0,1]$. Note that the log-determinant term is bounded as $\log(\det \widetilde{V}_T / \det \widetilde{V}_0) = \mathcal{O}(d \log T)$, and the noise-dependent terms are independent of $T$. We apply the lemma for two specific cases:

- **Case 1** ($p = 1/2$): Substituting $p = 1/2$ into Lemma B.1, the leading term scales with $T^{1-1/2}(\mathcal{O}(d\log T))^{1/2} = \widetilde{\mathcal{O}}(\sqrt{dT})$. Since the sum over a subset of indices $\{t : \omega_t = 1\}$ is bounded by the sum over all $t \in [T]$, we have:

$$
\sum_{t:\omega_t=1} \|\Delta z_{t,a_t,b_t}^*\|_{\widetilde{V}_{t-1}^{-1}} \leq \sum_{t=1}^{T} \|\Delta z_{t,a_t,b_t}^*\|_{\widetilde{V}_{t-1}^{-1}} = \widetilde{\mathcal{O}}(\sqrt{dT}).
\tag{B.19}
$$

  ignoring logarithm terms and omit sub-leading terms.

- **Case 2** ($p = 1$): Substituting $p = 1$, the leading term scales with $T^{1-1}(\mathcal{O}(d\log T))^1 = \widetilde{\mathcal{O}}(d)$. Similarly, bounding the partial sum by the total sum yields:

$$
\sum_{t:\omega_t<1} \|\Delta z_{t,a_t,b_t}^*\|_{\widetilde{V}_{t-1}^{-1}}^2 \leq \sum_{t=1}^{T} \|\Delta z_{t,a_t,b_t}^*\|_{\widetilde{V}_{t-1}^{-1}}^2 = \widetilde{\mathcal{O}}(d).
\tag{B.20}
$$

  ignoring logarithm terms and omit sub-leading terms.

Combining Equations (B.17) to (B.20) and recalling that $\beta_T = \widetilde{\mathcal{O}}(\sqrt{d} + \lambda M)$, the cumulative regret decomposes as:

$$
2R_T = I_1 + I_2,
$$

where:

$$
I_1 := C_0(L_\Theta + \beta_T)\sqrt{d_x}\log(T/\delta)\sum_{t=1}^{T} t^{-a} = \widetilde{\mathcal{O}}\big(\sqrt{d \cdot d_x}\, T^{1-\alpha}\big),
$$

$$
I_2 := 3\beta_T \sum_{t=1}^{T} \|\Delta z_{t,a_t,b_t}^*\|_{\widetilde{V}_{t-1}^{-1}} = \widetilde{\mathcal{O}}\big(d\sqrt{T} + d(\mathcal{C}+\mathcal{D})\big).
$$

Therefore, the total cumulative regret satisfies:

$$
\boxed{R_T = \widetilde{\mathcal{O}}\Big(\big(d + \sqrt{d \cdot d_x}\big)\sqrt{T} + d(\mathcal{C}+\mathcal{D})\Big).}
$$

This completes the proof of Theorem 5.2. □

## C. Proof of Theorem 5.3

We adopt the piecewise linear link function $\sigma(x) = \frac{1}{2} + x$ for $x \in [-\frac{1}{2}, \frac{1}{2}]$. For any scaling factor $\kappa > 0$, we define the scaled link function as $\sigma_\kappa(x) := \sigma(\kappa x)$. Throughout this analysis, we assume that post-serving contexts are absent. We first introduce a fundamental lower bound that encapsulates both statistical variance and adversarial corruptions.

**Lemma C.1** (Statistical and Corruption Lower Bound (Di et al., 2025, Theorem 5.4)). *For any learning algorithm $\mathcal{A}$, there exists an environment $\mathcal{I}_1$ such that the expected regret is lower-bounded as:*

$$\mathbb{E}[\text{Regret}(\mathcal{A}; \mathcal{I}_1)] \geq \Omega(d\sqrt{T} + dC),$$

*where $C$ denotes the total corruption budget.*

To further account for the impact of feedback latency, we characterize the fundamental limits under an adversarial delay model.

**Lemma C.2** (Adversarial Delay Lower Bound). *For any algorithm $\mathcal{A}$ operating within an adversarial delay budget $\Lambda$, there exists an instance $\mathcal{I}_2$ such that:*

$$\mathbb{E}[\text{Regret}(\mathcal{A}; \mathcal{I}_2)] \geq \Omega(\sqrt{d\Lambda}).$$

*Proof.* We adopt the continuous action set construction from Di et al. (2025). Consider a parameter $\Theta_* \in \{-\Delta, +\Delta\}^d$ where each coordinate $\Theta_{*,i}$ is drawn i.i.d. uniformly at random, with $\Delta = 1/8$. The action set is defined as $\mathcal{A} = \{x \in \mathbb{R}^d : \|x\|_2 \leq 1\}$, and the utility function is linear in the parameter, i.e., $u(a) = \langle \Theta_*, a \rangle$ for any action $a \in \mathcal{A}$. Under this construction, the optimal action is given by $a^* = \frac{1}{\sqrt{d}}\text{sign}(\Theta_*)$, yielding an optimal utility of

$$u(a^*) = \left\langle \Theta_*, \frac{1}{\sqrt{d}}\text{sign}(\Theta_*) \right\rangle = \frac{1}{\sqrt{d}} \sum_{i=1}^{d} |\Theta_{*,i}| = \frac{1}{\sqrt{d}} \cdot d\Delta = \sqrt{d}\Delta. \tag{C.21}$$

We now describe the adversary's delay assignment strategy. Define

$$M = \left\lfloor \frac{-1 + \sqrt{1 + 8\Lambda}}{2} \right\rfloor = \Theta(\sqrt{\Lambda}), \tag{C.22}$$

which corresponds to the largest integer satisfying $\frac{M(M+1)}{2} \leq \Lambda$. The adversary assigns delay $\tau_t = M - t + 1$ to round $t$ for all $t \in [M]$, ensuring that all feedback is withheld until after round $M$. This creates a blind phase during which the learner receives no information.

It remains to analyze the regret incurred during this blind phase. In the dueling bandits framework, the learner selects two actions $(a_t, b_t)$ at each round, and the per-round regret is defined as

$$\text{Regret}_t = \left(u(a^*) - u(a_t)\right) + \left(u(a^*) - u(b_t)\right), \tag{C.23}$$

which measures the suboptimality of both selected actions relative to the optimal action $a^*$.

For each round $t \in [M]$, the learner operates with an empty history $\mathcal{H}_t = \emptyset$. Since the learner has received no feedback, the actions $a_t$ and $b_t$ are chosen independently of the true parameter $\Theta_*$. By the symmetry of the prior distribution, each coordinate satisfies $\mathbb{E}[\Theta_{*,i}] = \frac{1}{2}(+\Delta) + \frac{1}{2}(-\Delta) = 0$, which implies $\mathbb{E}[\Theta_*] = \mathbf{0}$. Consequently, for any action $a_t$ that is independent of $\Theta_*$, we have

$$\mathbb{E}[u(a_t)] = \mathbb{E}[\langle \Theta_*, a_t \rangle] = \langle \mathbb{E}[\Theta_*], a_t \rangle = \langle \mathbf{0}, a_t \rangle = 0. \tag{C.24}$$

By the same argument, $\mathbb{E}[u(b_t)] = 0$. Therefore, the expected per-round regret is

$$\mathbb{E}[\text{Regret}_t] = 2u(a^*) - \mathbb{E}[u(a_t)] - \mathbb{E}[u(b_t)] = 2\sqrt{d}\Delta - 0 - 0 = 2\sqrt{d}\Delta. \tag{C.25}$$

Aggregating over all $M$ rounds in the blind phase yields

$$\mathbb{E}[\text{Regret}] \geq \sum_{t=1}^{M} \mathbb{E}[\text{Regret}_t] = M \cdot 2\sqrt{d}\Delta = \Omega(\sqrt{\Lambda} \cdot \sqrt{d}) = \Omega(\sqrt{d\Lambda}), \tag{C.26}$$

which completes the proof. $\qquad\square$

**Lemma C.3** (Stochastic Delay Lower Bound). *For any algorithm $\mathcal{A}$ with stochastic delays of mean $\mu_\tau$, there exists an instance $\mathcal{I}_3$ such that*

$$\mathbb{E}[\mathrm{Regret}(\mathcal{A}; \mathcal{I}_3)] \geq \Omega(d\mu_\tau). \tag{C.27}$$

*Proof.* Consider the deterministic delay distribution $\tau_t = \mu_\tau$ for all $t$, which trivially satisfies $\mathbb{E}[\tau] = \mu_\tau$. Let $\Theta_* \in \{-\Delta, +\Delta\}^d$ where each coordinate $\Theta_{*,i}$ is drawn i.i.d. uniformly at random, with $\Delta = 1/4$. The action set is defined as the hypercube $\mathcal{A} = \{-1, 1\}^d$, and the utility function is linear in the parameter, i.e., $u(a) = \langle \Theta_*, a \rangle = \sum_{i=1}^d a_i \Theta_{*,i}$ for any action $a \in \mathcal{A}$. Under this construction, the optimal action is $a^* = \mathrm{sign}(\Theta_*)$, which matches the sign of each coordinate precisely, yielding an optimal utility of

$$u(a^*) = \sum_{i=1}^d \mathrm{sign}(\Theta_{*,i}) \cdot \Theta_{*,i} = \sum_{i=1}^d |\Theta_{*,i}| = d\Delta. \tag{C.28}$$

For all rounds $t \leq \mu_\tau$, the learner operates with an empty history $\mathcal{H}_t = \emptyset$, creating a blind phase during which no feedback is available. Since the learner has received no information, the actions $a_t$ and $b_t$ are chosen independently of the true parameter $\Theta_*$. By the symmetry of the prior distribution, each coordinate satisfies $\mathbb{E}[\Theta_{*,i}] = \frac{1}{2}(+\Delta) + \frac{1}{2}(-\Delta) = 0$, which implies $\mathbb{E}[\Theta_*] = \mathbf{0}$. Consequently, for any action $a_t$ that is independent of $\Theta_*$, we have

$$\mathbb{E}[u(a_t)] = \mathbb{E}[\langle \Theta_*, a_t \rangle] = \langle \mathbb{E}[\Theta_*], a_t \rangle = \langle \mathbf{0}, a_t \rangle = 0. \tag{C.29}$$

By the same argument, $\mathbb{E}[u(b_t)] = 0$. In the dueling bandits framework, the learner selects two actions $(a_t, b_t)$ at each round, and the expected per-round regret is

$$\mathbb{E}[\mathrm{Regret}_t] = 2u(a^*) - \mathbb{E}[u(a_t)] - \mathbb{E}[u(b_t)] = 2d\Delta - 0 - 0 = 2d\Delta. \tag{C.30}$$

Aggregating over all $\mu_\tau$ rounds in the blind phase yields

$$\mathbb{E}[\mathrm{Regret}] \geq \sum_{t=1}^{\mu_\tau} \mathbb{E}[\mathrm{Regret}_t] = \mu_\tau \cdot 2d\Delta = \Omega(d\mu_\tau), \tag{C.31}$$

which completes the proof. $\square$

*Proof of Theorem 5.3.* By Lemmas C.1–C.3, for any algorithm $\mathcal{A}$, we have

$$\sup_{\mathcal{I}} \mathbb{E}[\mathrm{Regret}(\mathcal{A}; \mathcal{I})] \geq \max \left\{ \Omega(d\sqrt{T} + dC), \Omega(\sqrt{d\Lambda}), \Omega(d\mu_\tau) \right\}. \tag{C.32}$$

Since adversarial and stochastic delays are mutually exclusive, and using the inequality $\max(A, B) \geq (A + B)/2$, we obtain

$$\sup_{\mathcal{I}} \mathbb{E}[\mathrm{Regret}(\mathcal{A}; \mathcal{I})] \geq \Omega \left( d\sqrt{T} + dC + \max\{\sqrt{d\Lambda}, d\mu_\tau\} \right). \tag{C.33}$$

Finally, for any $\kappa > 0$, consider the scaled link function $\sigma_\kappa(x) = \sigma(\kappa x)$. By the same scaling argument as in Di et al. (2025, Theorem 5.4), the lower bound becomes $\Omega \left( (d\sqrt{T} + dC + \max\{\sqrt{d\Lambda}, d\mu_\tau\})/\kappa \right)$. $\square$

*Remark C.4* (Different Constructions). The adversarial delay bound uses the continuous action set $\{x : \|x\|_2 \leq 1\}$, while the stochastic delay bound uses the discrete set $\{e_i\}_{i=1}^d$. This is because:

- Adversarial delays can concentrate budget to create a blind phase of length $\Theta(\sqrt{\Lambda})$, and the continuous set yields per-round regret $\Theta(\sqrt{d})$, giving $\Omega(\sqrt{d\Lambda})$.

- Stochastic delays create a fixed blind phase of length $\mu_\tau$, and the discrete set with $d$ sub-instances yields $\Omega(d\mu_\tau)$ via minimax over coordinates.

Since these are different instances combined via minimax, both bounds are valid.

*Remark* C.5 (On the Nature of the Lower Bound). Our lower bound is established via the standard minimax framework, where different terms may arise from different hard instances. This is consistent with prior works (Di et al., 2025; He et al., 2022; Gupta et al., 2019). The additive form $\Omega(A + B + C)$ follows from $\max(A, B, C) \geq (A + B + C)/3$, which is the standard way to express minimax lower bounds in the bandit literature.

*Remark* C.6 (On the Additive Form). The lower bound is established via the standard minimax framework following Di et al. (2025); He et al. (2022); Gupta et al. (2019). Different terms arise from different hard instance families:

- $\Omega(d\sqrt{T}/\kappa)$: Statistical complexity from continuous action set with $\Delta = \Theta(\sqrt{d/T})$

- $\Omega(dC/\kappa)$: Corruption cost from discrete action set with $d$ sub-instances

- $\Omega(\sqrt{d\Lambda}/\kappa)$ or $\Omega(d\mu_\tau/\kappa)$: Delay cost from blind phase construction

# D. Experimental Details

## D.1. Implementation Details

*Table D.1.* Summary of Experimental Settings and Hyperparameters.

| Parameter | Symbol | Value |
|---|:---:|:---:|
| *Environment* | | |
| Time Horizon | $T$ | 2,000 |
| Pre-serving Dimension | $d_x$ | 10 |
| Number of Arms | $K$ | 10 |
| Number of Runs | - | 10 |
| *Robustness Settings* | | |
| Corruption Budget | $\mathcal{C}$ | 25 |
| Strategic Delay Budget | $\Lambda$ | 10,000 |
| Stochastic Delay Mean | $\mu_\tau$ | 100 |
| Stochastic Delay Std | $\sigma$ | 100 |
| *Algorithm Hyperparameters* | | |
| Regularization | $\lambda$ | 1.0 |
| MLP Hidden Units | - | (64, 64) |
| MLP Learning Rate | $\eta$ | $10^{-3}$ |
| Optimizer | Adam (Kingma, 2014) | |

In Section 6, we empirically evaluate the performance of our proposed method, RCDP-UCB, alongside baseline algorithms within a synthetic contextual dueling bandit framework. The dimension of the pre-serving context was set to $d_x = 10$. At each round $t$, the environment generates a pool of $K = 10$ arms. The ground-truth parameters $\Theta^* \in \mathbb{R}^d$ were sampled from a standard normal distribution and normalized to ensure valid utility scales. To investigate the method's adaptability to complex environmental dynamics, we tested four distinct mapping functions $\phi : \mathcal{X} \to \mathcal{Y}$ governing the relationship between pre-serving and post-serving contexts: *Polynomial*, *Absolute*, *Cosine*, and *Sinusoidal*. To verify the robustness of our method against data irregularities, we introduced both adversarial corruptions and feedback delays. The environment was constrained by a corruption budget of $\mathcal{C} = 25$ and a strategic delay budget of $\Lambda = 10,000$. Additionally, stochastic delays were modeled with a mean of $\mu_\tau = 100$ and a standard deviation of $\sigma = 100$. For the neural network approximation, we employed a Multi-Layer Perceptron (MLP) consisting of two hidden layers with 64 units each. The network was optimized using Adam (Kingma, 2014) with a learning rate of $\eta = 10^{-3}$ and a regularization coefficient of $\lambda = 1.0$. The experiments were conducted over a time horizon of $T = 2,000$ rounds, and all reported results were averaged over 10 independent runs with different random seeds to ensure statistical reliability. For our method, we approximated the unknown mapping $f$ using a Multi-Layer Perceptron (MLP) with two hidden layers of 64 units each and ReLU activations. The network was optimized using Adam with a learning rate of $10^{-3}$ and trained for 2 epochs per update. Regarding the bandit hyperparameters, we set the regularization parameter to $\lambda = 1.0$.

A comprehensive summary of the experimental configurations and hyperparameters is provided in Table D.1.

All experiments were conducted on a workstation equipped with an AMD Ryzen 9 9950X3D 16-Core Processor and 48GB of RAM. The algorithms were implemented in Python 3.10, utilizing PyTorch for the neural network components and NumPy for vector operations. The experiments were executed in a CPU-only environment, as the computational overhead of the multi-layer perceptron (MLP) utilized for context mapping was negligible, and no GPU acceleration was required.

### D.2. Related Algorithms for Contextual Dueling Bandits

We review four representative algorithms for the contextual dueling bandit problem: RCDB, COLSTIM, MaxInP, and MaxPair-UCB. All these algorithms assume a linear relationship between the context-action features and the latent utility or reward.

**RCDB (Robust Contextual Dueling Bandits).** Proposed by (Di et al., 2025), RCDB is designed to handle adversarial feedback where a strong adversary may flip the preference labels. To mitigate the impact of malicious feedback, RCDB employs an *uncertainty-weighted Maximum Likelihood Estimator (MLE)*. The estimator $\hat{\theta}_t$ is obtained by solving the weighted score equation:

$$\lambda\theta + \sum_{\tau=1}^{t-1} w_\tau(\sigma(\phi_\tau^\top\theta) - o_\tau)\phi_\tau = 0, \tag{D.34}$$

where $\phi_\tau = \phi(x_\tau, a_\tau) - \phi(x_\tau, b_\tau)$ denotes the differential feature vector, and $o_\tau$ is the observed feedback. The key innovation lies in the uncertainty-dependent weights $w_\tau = \min\{1, \alpha/\|\phi_\tau\|_{\Sigma_\tau^{-1}}\}$, which down-weight samples with high uncertainty to limit the adversary's influence. The algorithm selects the arm pair $(a_t, b_t)$ that maximizes a robust Upper Confidence Bound (UCB) objective:

$$(a_t, b_t) = \underset{a,b \in \mathcal{A}_t}{\operatorname{argmax}} \left\{ (\phi(x_t, a) + \phi(x_t, b))^\top\hat{\theta}_t + \beta_t\|\phi(x_t, a) - \phi(x_t, b)\|_{\Sigma_t^{-1}} \right\}. \tag{D.35}$$

**COLSTIM (CoLST Imitator).** Introduced by (Bengs et al., 2022), COLSTIM operates under the Linear Stochastic Transitivity (LST) model. It adopts a randomized exploration strategy that imitates the underlying feedback process. Specifically, at each round $t$, it generates perturbed utility estimates for each arm $a$ by adding noise scaled by the confidence width:

$$\widetilde{u}_{t,a} = x_{t,a}^\top\hat{\theta}_t + \epsilon_{t,a}\|x_{t,a}\|_{M_t^{-1}}, \tag{D.36}$$

where $\epsilon_{t,a}$ is a random perturbation sampled from a specific distribution (e.g., Gumbel) and $M_t$ is the Gram matrix. The first arm $a_t$ is chosen to maximize this perturbed utility, $\widetilde{u}_{t,a}$. The second arm $b_t$ is then selected as the *toughest competitor* to $a_t$, maximizing the optimistic pairwise difference:

$$b_t = \underset{b \in \mathcal{A}_t}{\operatorname{argmax}} \left( (x_{t,b} - x_{t,a_t})^\top\hat{\theta}_t + c_t\|x_{t,b} - x_{t,a_t}\|_{M_t^{-1}} \right). \tag{D.37}$$

**MaxInP (Maximum Informative Pair).** Developed by (Saha, 2021), MaxInP focuses on efficient exploration by maintaining a set of "promising" arms, $\mathcal{C}_t$, which contains arms that are plausibly the best with high probability. The algorithm avoids playing suboptimal arms by restricting selection to $\mathcal{C}_t$. Within this set, it selects the pair $(a_t, b_t)$ that maximizes the information gain, quantified by the variance of the pairwise difference:

$$(a_t, b_t) = \underset{a,b \in \mathcal{C}_t}{\operatorname{argmax}} \|\phi(x_t, a) - \phi(x_t, b)\|_{V_t^{-1}}, \tag{D.38}$$

where $V_t$ is the covariance matrix of the differential features. This *max-variance* strategy ensures that the algorithm continually reduces uncertainty in the directions most relevant to distinguishing the best arms.

**MaxPair-UCB.** As discussed in (Di et al., 2023), MaxPair-UCB is a UCB-based algorithm that simplifies the selection process of MaxInP by eliminating the explicit construction of a promising set. Instead, it directly selects the pair that maximizes an optimistic estimate of the sum-reward combined with an exploration bonus on the pairwise difference. The selection rule is given by:

$$(a_t, b_t) = \underset{a,b \in \mathcal{A}_t}{\operatorname{argmax}} \left\{ (x_{t,a} + x_{t,b})^\top\hat{\theta}_t + \beta_t\|x_{t,a} - x_{t,b}\|_{\Sigma_t^{-1}} \right\}. \tag{D.39}$$

This approach naturally balances exploitation (maximizing the estimated sum score $(x_{t,a} + x_{t,b})^\top \hat{\theta}_t$) and exploration (maximizing the pairwise uncertainty $\|x_{t,a} - x_{t,b}\|_{\Sigma_t^{-1}}$), and serves as a strong baseline in variance-aware and adversarial settings.

### D.3. Implementation with PyTorch.

Listing 1 demonstrates a simplified PyTorch implementation of our proposed algorithm, `RCDP-UCB`. The class integrates the online learning of the post-serving context mapping with the robust bandit strategy. To approximate the unknown mapping, a neural network is employed and trained online via the `train_mapping` method, which updates the model using collected context-outcome pairs. The `get_features` method then concatenates the original context with the network's prediction to form the augmented feature vector used for decision-making. In the `select_arms` method, we employ an adaptive UCB strategy that relies on the *Observed History* matrix (`self.W_inv`) to construct confidence bounds based on actual localized information. Critically, robustness is enforced by the `get_weight` method, which computes a down-weighting factor using the *Full History* matrix (`self.V_inv`). This ensures that updates with high forward-looking uncertainty are throttled before they can destabilize the model in the `update` step, effectively neutralizing the impact of potential strategic delays or corruptions.

```python
class RCDP_UCB(nn.Module):
    def __init__(self, dim, alpha=0.5, lambda_reg=1.0, rho=1.0, kappa=0.1):
        super().__init__()
        self.kappa = kappa
        self.dim = dim
        self.theta_hat = torch.zeros(dim)

        # Full History Matrix V (for Selection & Weighting)
        self.V_inv = (1.0 / lambda_reg) * torch.eye(dim)

        # Observed History Matrix W (for Gradient Update)
        self.W_inv = (1.0 / lambda_reg) * torch.eye(dim)

        self.alpha = alpha
        self.rho = rho # Robustness parameter (sqrt(d)/C_eff)

    def select_arms(self, Z):
        """
        Z: Features including predicted post-serving context (K, dim)
        Returns: a_t, b_t, omega_t
        """
        # 1. Champion (Greedy)
        utilities = Z @ self.theta_hat
        a_t = torch.argmax(utilities)
        z_a = Z[a_t]

        # 2. Challenger (Use V_inv for Optimism in Face of Uncertainty)
        Delta_Z = Z - z_a
        mean_diff = Delta_Z @ self.theta_hat

        # KEY: Use V_inv (Full History) for Confidence Width
        width_V = torch.sqrt(torch.sum((Delta_Z @ self.V_inv) * Delta_Z, dim=1))

        scores_b = mean_diff + self.alpha * width_V
        b_t = torch.argmax(scores_b)

        # 3. Calculate Robust Weight (omega_t)
        delta_z = Z[a_t] - Z[b_t]
        norm_V = torch.sqrt(delta_z @ self.V_inv @ delta_z)

        if norm_V < 1e-9:
            omega_t = 1.0
        else:
            omega_t = min(1.0, self.rho / norm_V)
```

```
            # 4. Phantom Update of V (Full History)
            # We update V immediately to reflect the "commitment" to this arm pair
            # Formula: V <- V + kappa * omega_t * outer(delta, delta)
            # Using Sherman-Morrison for V_inv
            v_vec = torch.sqrt(self.kappa * omega_t) * delta_z
            Av = self.V_inv @ v_vec
            denom = 1 + torch.dot(v_vec, Av)
            self.V_inv -= torch.outer(Av, Av) / denom

            return a_t, b_t, omega_t

    def update(self, delta_z, outcome, omega_t):
        """
        Real Update when Feedback Arrives
        """
        # 1. Update Observed Matrix W using Robust Weight
        w_vec = torch.sqrt(self.kappa * omega_t) * delta_z
        Aw = self.W_inv @ w_vec
        denom_w = 1 + torch.dot(w_vec, Aw)
        self.W_inv -= torch.outer(Aw, Aw) / denom_w

        # 2. Weighted Gradient Step
        mu_val = torch.sigmoid(torch.dot(self.theta_hat, delta_z))

        # Gradient direction weighted by omega_t * W_inv
        step = self.W_inv @ (omega_t * (outcome - mu_val) * delta_z)
        self.theta_hat += step
```

*Listing 1.* PyTorch Implementation of RCDP-UCB (Simplified)

# E. Additional Experiments

## E.1. Experiments on Real-world Datasets

We validate the robustness and scalability of RCDP-UCB using eight diverse classification datasets from the UCI Machine Learning Repository and OpenML. For each dataset, we partition the feature vector into visible pre-serving contexts ($\mathbf{x}_t$) and latent post-serving features ($\mathbf{y}_t$), as summarized in Table E.2.

**Dataset Preparation.** In each round $t$, the environment randomly samples $K = 30$ instances from the dataset to form the available set of arms $\mathcal{A}_t$. The ground-truth utility of an arm is defined by its class label $c \in \{0, \ldots, C - 1\}$, where higher-indexed classes are considered preferable. All feature values are standardized using zero-mean and unit variance before the start of the simulation.

**Empirical Results.** Figure E.1 shows the results for three representative datasets: Magic Gamma Telescope ($d = 6$), Statlog ($d = 20$), and Spambase ($d = 30$). Across all datasets and both delay regimes (strategic starvation and stochastic network latency), RCDP-UCB consistently outperforms robust and non-robust baselines. Notably, even on real-world distributions where the linear preference assumption is only an approximation, RCDP-UCB's weighting mechanism effectively mitigates the bias introduced by the combination of adversarial corruption and delayed feedback.

*Table E.2.* Statistics of real-world datasets used in the experiments. The feature vectors are split into pre-serving $d$-dim and post-serving $e$-dim components. Utility is defined by class labels.

| Dataset | Samples | Total Features | Pre ($d$) | Post ($e$) | Classes |
|---|---|---|---|---|---|
| Magic | 19,020 | 10 | 6 | 4 | 2 |
| Statlog (Satimage) | 6,430 | 36 | 20 | 16 | 6 |
| Spambase | 4,601 | 57 | 30 | 27 | 2 |

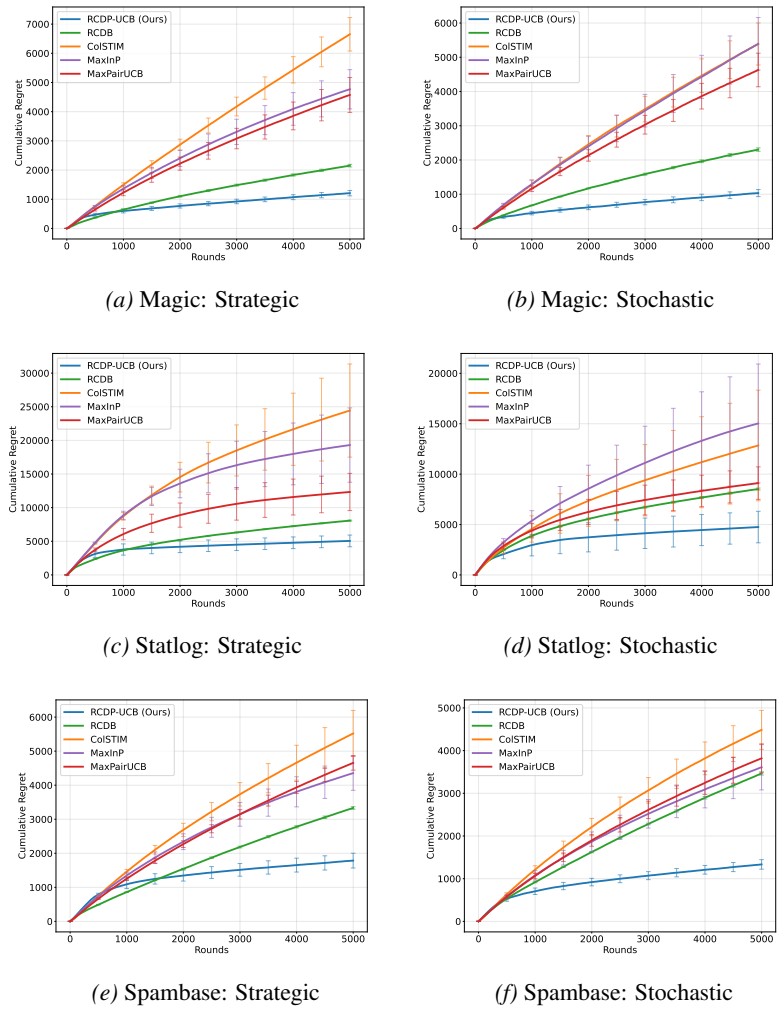

*(a)* Magic: Strategic

*(b)* Magic: Stochastic

*(c)* Statlog: Strategic

*(d)* Statlog: Stochastic

*(e)* Spambase: Strategic

*(f)* Spambase: Stochastic

*Figure E.1.* **Real-world Dataset Results**: Cumulative regret comparison on Magic, Statlog, and Spambase datasets under adversarial corruption ($\mathcal{C} = 25$), strategic delays ($\Lambda = 10,000$), and stochastic delays ($\mu_\tau = 100$). All experiments were conducted across 10 runs with random seeds.

## E.2. Increasing Context Dimension

In this section, we provide an extensive empirical evaluation of RCDP-UCB across varying pre-serving context dimensions $d_x \in \{10, 20, 30\}$. We investigate two scenarios: with and without post-serving learning.

**Sensitivity without Baseline Post-Serving Learning.** Figure E.2 illustrates the performance scaling with respect to dimension $d$ in the linear task using only pre-serving features. RCDP-UCB maintains robust sublinear regret across all dimensions, with the performance gap relative to RCDB widening as $d$ increases. This validates the efficacy of our weighting mechanism in high-dimensional settings under joint $\mathcal{C} + \mathcal{D}$ perturbations.

**Sensitivity with Baseline Post-Serving Learning.** We further evaluate robustness across all three mappings (Absolute, Polynomial, and Sinusoidal) when all baselines are equipped with MLP-based post-serving learning (Figure E.3, Figure E.4, Figure E.5). Even with the additional information provided by $\hat{\phi}_t$, standard baselines suffer from the unified impact of corruption and delay. RCDP-UCB consistently maintains superior performance, demonstrating that its advantage stems from the adaptive information-weighted estimation rather than mere feature engineering. The stability across Absolute, Polynomial, and Sinusoidal tasks confirms the algorithm's versatility in handling diverse latent dynamics as the complexity of the feature manifold increases.

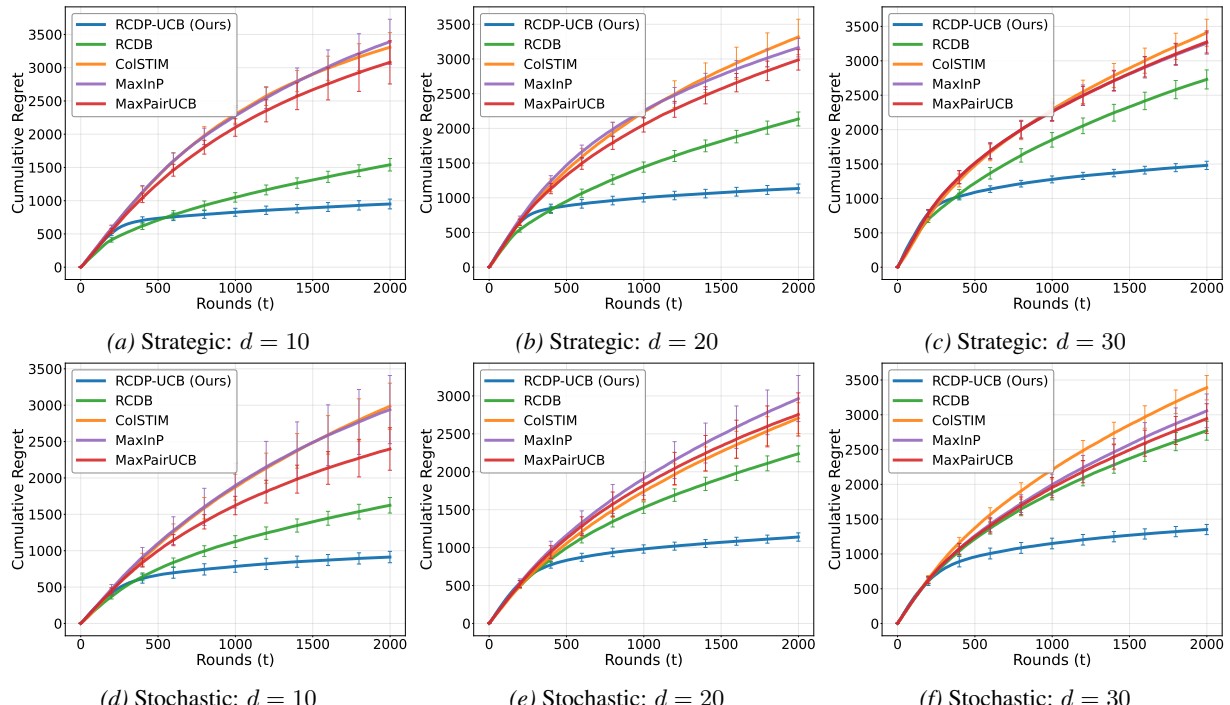

*Figure E.2.* **Linear Problem (Pre-serving Baselines)**: Cumulative regret with increasing context dimension $d$ under adversarial corruption ($\mathcal{C} = 25$), strategic delays ($\Lambda = 10,000$), and stochastic delays ($\mu_\tau = 100$). All experiments were conducted across 10 runs with random seeds.

### E.3. Increasing Action Dimension

We examine the scalability of RCDP-UCB as the number of available arms $K$ increases from 10 to 30. Similar to the context dimension analysis, we consider both pre-serving and post-serving baseline scenarios.

**Sensitivity without Baseline Post-Serving Learning.** Figure E.6 illustrates the performance scaling with respect to the number of arms $K$ in the linear task. RCDP-UCB maintains stable sublinear regret as $K$ increases, consistently outperforming RCDB and other competitive baselines. These results underscore the robustness of our proposed framework under concurrent adversarial corruptions and feedback delays.

**Sensitivity with Baseline Post-Serving Learning.** The robustness of RCDP-UCB is further validated across Absolute, Polynomial, and Sinusoidal mappings when all baselines incorporate post-serving learning (Figure E.7, Figure E.8, Figure E.9). Even with the ability to predict latent contexts, baselines struggle to maintain low regret as the action space expands. In contrast, the performance of RCDP-UCB remains remarkably consistent. This suggests that the adaptive weighting mechanism effectively controls the variance of the MLE estimation, preventing the combined noise of corruption and delay from overwhelming the identification of the optimal arm even as the competition among arms increases.

### E.4. Varying Corruption Budget

We investigate the robustness of RCDP-UCB as the total budget of adversarial corruption $\mathcal{C}$ varies. Specifically, we test $\mathcal{C} \in \{10, 15, 20\}$ under a fixed mean stochastic delay $\mu_\tau = 100$, standard deviation $\sigma = 10$.

**Empirical Analysis.** Figure E.10 shows the cumulative regret for the linear problem with context dimensions $d = 10$ and $d = 20$. As the corruption budget increases, the bias introduced by the adversary becomes more significant. However, RCDP-UCB consistently maintains lower regret and faster convergence compared to both robust and non-robust baselines. This is achieved through the weighted MLE framework, where the influence of potentially corrupted observations is balanced by the information gain weights $\omega_t$ and the corruption-aware parameter estimation, ensuring that the learner remains effective even when a portion of the preferences are manipulated.

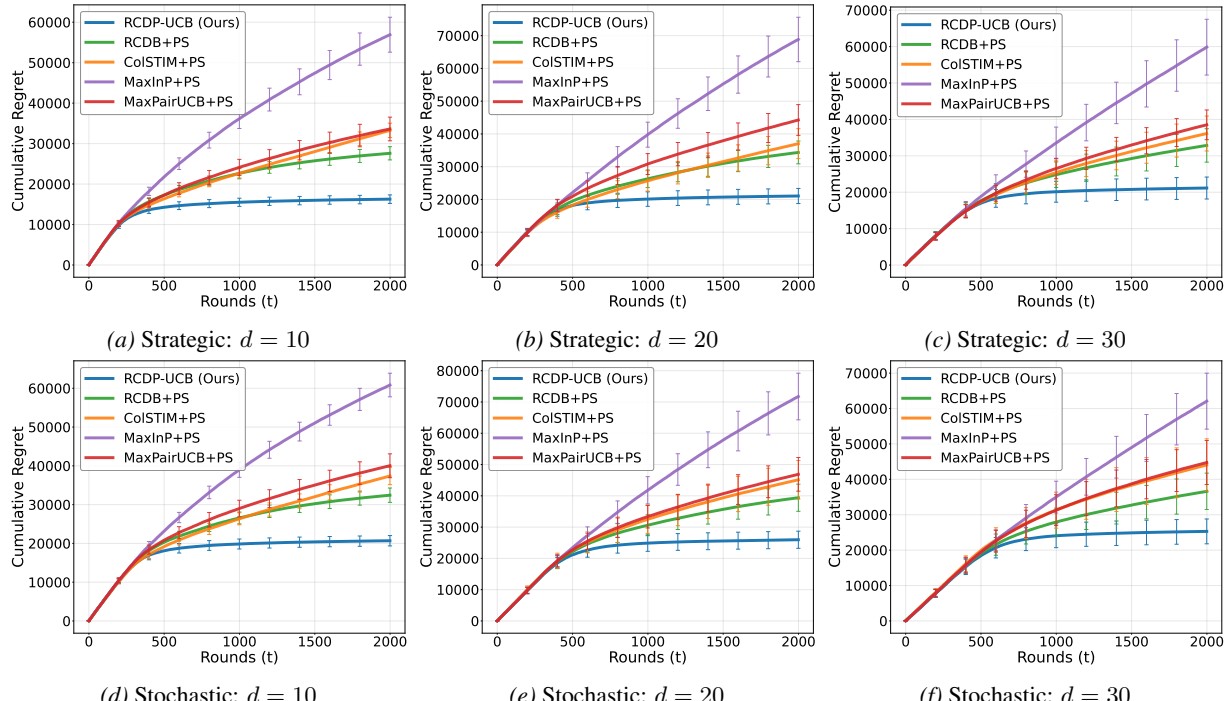

*Figure E.3.* **Absolute Mapping (Post-serving Learning)**: Cumulative regret with increasing context dimension $d$ under adversarial corruption ($\mathcal{C} = 25$), strategic delays ($\Lambda = 10,000$), and stochastic delays ($\mu_\tau = 100$), where all baselines learn $\phi_*$. All experiments were conducted across 10 runs with random seeds.

### E.5. Varying Total Budget of Adversarial Delay and Mean of Sub-Gaussian Delay

We analyze the performance of RCDP-UCB as the severity of feedback delays increases. Specifically, we vary the strategic delay budget $\Lambda$ from 3,600 to 10,000 and the mean of stochastic sub-Gaussian delays $\mu_\tau$ from 60 to 100.

**Sensitivity without Baseline Post-Serving Learning.** Figure E.11 summarizes the cumulative regret for the linear task. As the delay budget or mean delay increases, the learner receives feedback less frequently, leading to slower parameter convergence. RCDP-UCB demonstrates superior robustness compared to robust baselines like RCDB, maintaining lower regret even under severe starvation attacks.

**Sensitivity with Baseline Post-Serving Learning.** The robustness is further evaluated across Absolute, Polynomial, and Sinusoidal mappings where all baselines are augmented with post-serving learning (Figure E.12, Figure E.13, Figure E.14). RCDP-UCB consistently outperforms the baselines across all delay levels and mappings. The adaptive weighting mechanism in RCDP-UCB ensures that even when delays are large, the parameter updates are properly normalized by the local information density (captured by $\omega_t$), allowing for robust learning where standard MLE-based approaches struggle.

### E.6. Increasing STD. of Sub-Gaussian Delay

We assess the sensitivity of RCDP-UCB to the variance of stochastic delays by varying $\sigma$ from 0 to $\sqrt{150}$, while fixing the mean delay $\mu_\tau = 100$ and corruption budget $\mathcal{C} = 25$. Figure E.15 illustrates that the cumulative regret of RCDP-UCB remains remarkably stable across all evaluated noise levels. This empirical evidence validates that RCDP-UCB is robust to both the expected magnitude and the stochastic fluctuations of delays, a result consistent with the theoretical bounds established in Theorems 5.2 and 5.3.

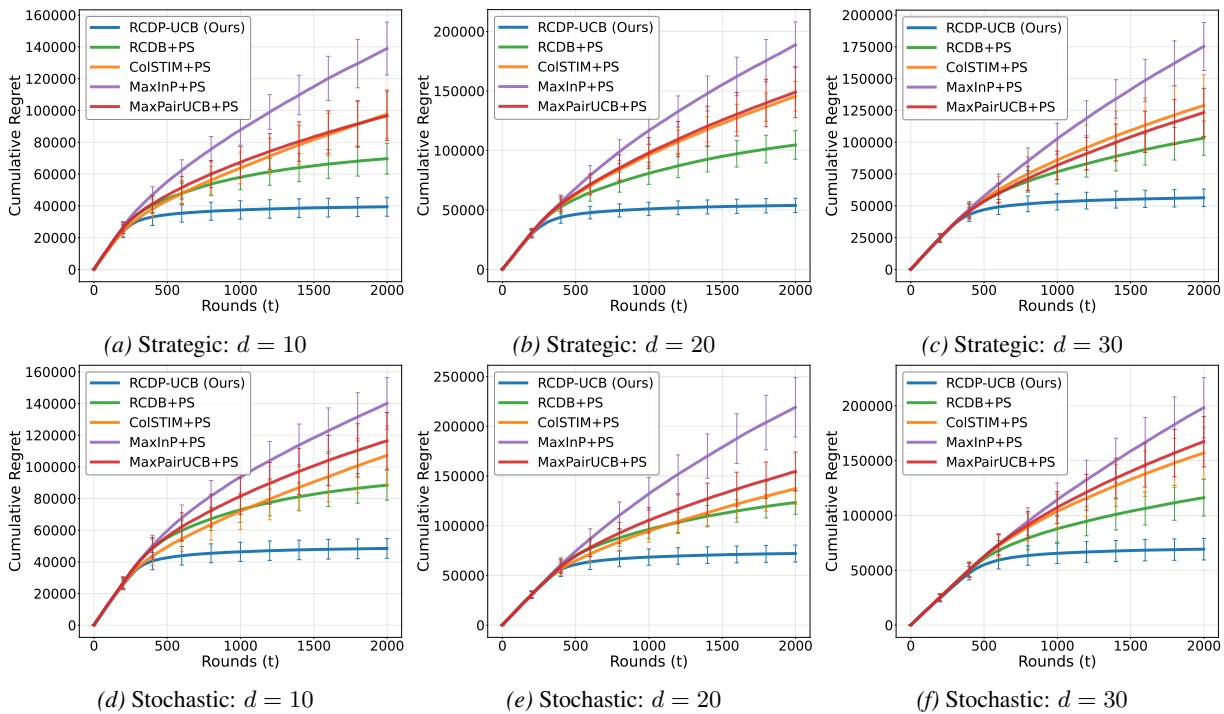

*Figure E.4.* **Polynomial Mapping (Post-serving Learning)**: Cumulative regret with increasing context dimension $d$ under adversarial corruption ($\mathcal{C} = 25$), strategic delays ($\Lambda = 10,000$), and stochastic delays ($\mu_\tau = 100$), where all baselines learn $\phi_*$. All experiments were conducted across 10 runs with random seeds.

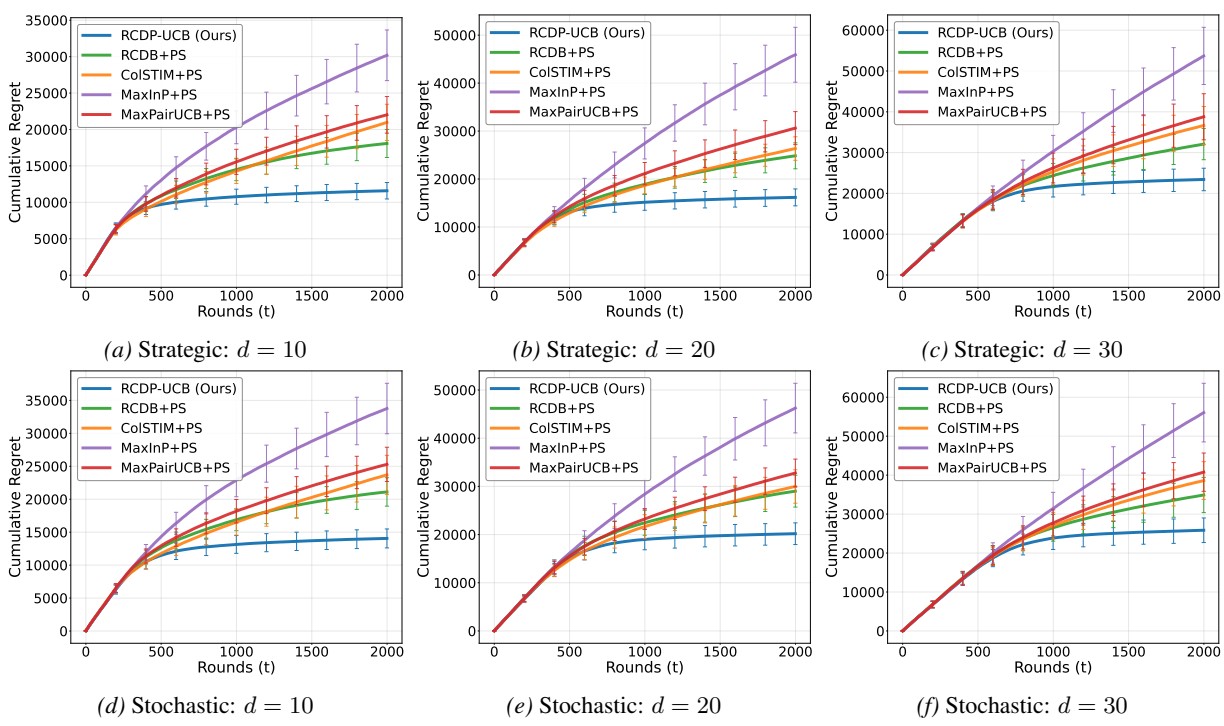

*Figure E.5.* **Sinusoidal Mapping (Post-serving Learning)**: Cumulative regret with increasing context dimension $d$ under adversarial corruption ($\mathcal{C} = 25$), strategic delays ($\Lambda = 10,000$), and stochastic delays ($\mu_\tau = 100$), where all baselines learn $\phi_*$. All experiments were conducted across 10 runs with random seeds.

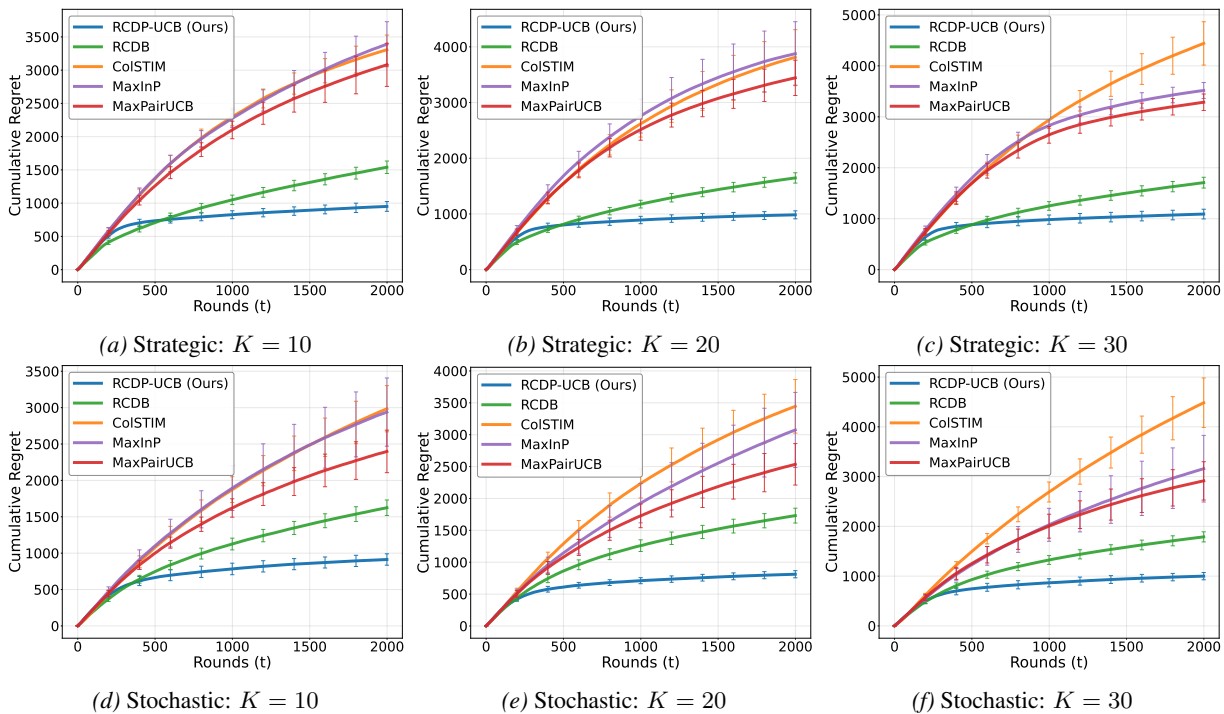

*Figure E.6.* **Polynomial Mapping (Pre-serving Baselines)**: Cumulative regret with increasing number of arms $K$ under adversarial corruption ($\mathcal{C} = 25$), strategic delays ($\Lambda = 10,000$), and stochastic delays ($\mu_\tau = 100$). All experiments were conducted across 10 runs with random seeds.

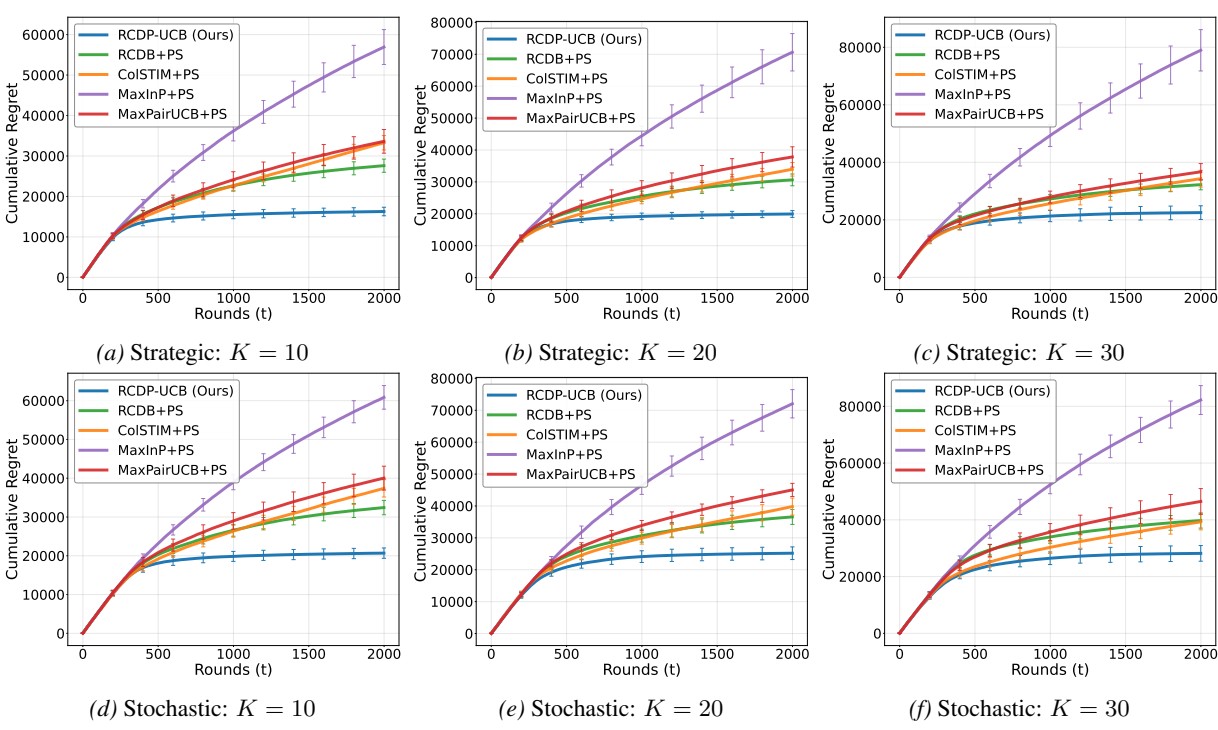

*Figure E.7.* **Absolute Mapping (Post-serving Learning)**: Cumulative regret with increasing number of arms $K$ under adversarial corruption ($\mathcal{C} = 25$), strategic delays ($\Lambda = 10,000$), and stochastic delays ($\mu_\tau = 100$), where all baselines learn $\phi_*$. All experiments were conducted across 10 runs with random seeds.

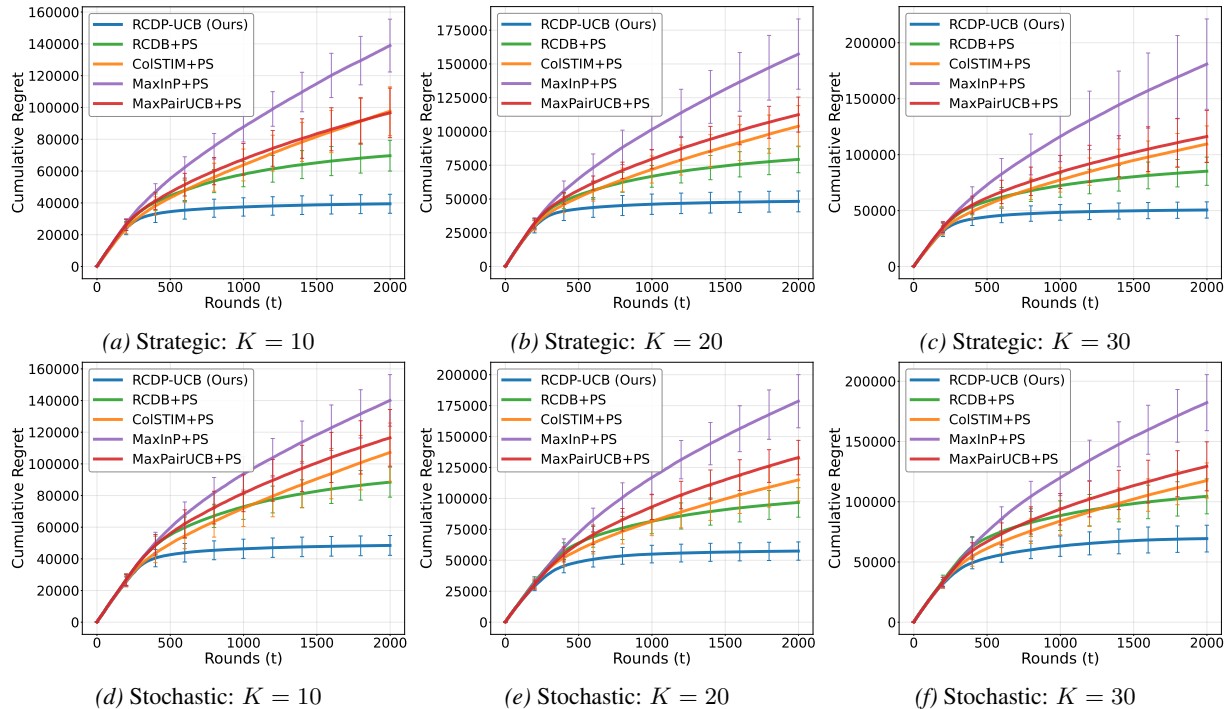

*Figure E.8.* **Polynomial Mapping (Post-serving Learning)**: Cumulative regret with increasing number of arms $K$ under adversarial corruption ($\mathcal{C} = 25$), strategic delays ($\Lambda = 10,000$), and stochastic delays ($\mu_\tau = 100$), where all baselines learn $\phi_*$. All experiments were conducted across 10 runs with random seeds.

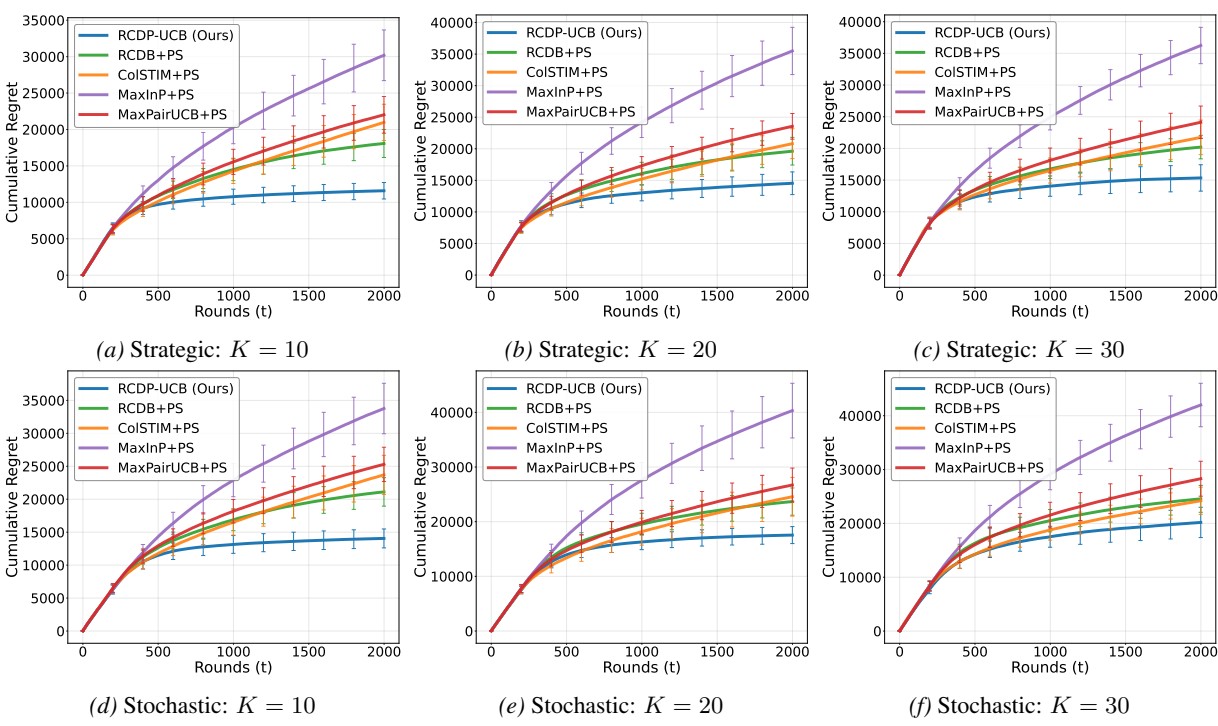

*Figure E.9.* **Sinusoidal Mapping (Post-serving Learning)**: Cumulative regret with increasing number of arms $K$ under adversarial corruption ($\mathcal{C} = 25$), strategic delays ($\Lambda = 10,000$), and stochastic delays ($\mu_\tau = 100$), where all baselines learn $\phi_*$. All experiments were conducted across 10 runs with random seeds.

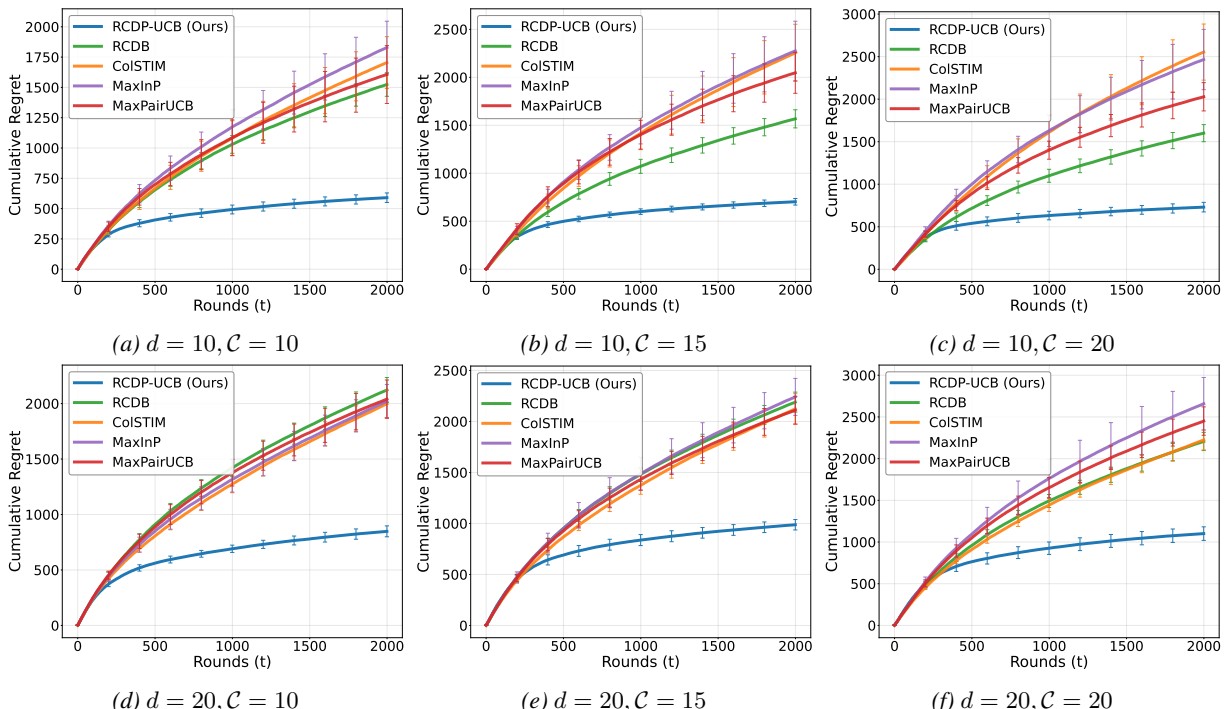

*Figure E.10.* **Polynomial Mapping (Varying Corruption)**: Cumulative regret with varying corruption budget $\mathcal{C}$ for different context dimensions ($d = 10, 20$). All experiments were conducted under fixed stochastic delay parameters ($\mu_\tau = 100, \sigma = 100$) and averaged over 10 runs.

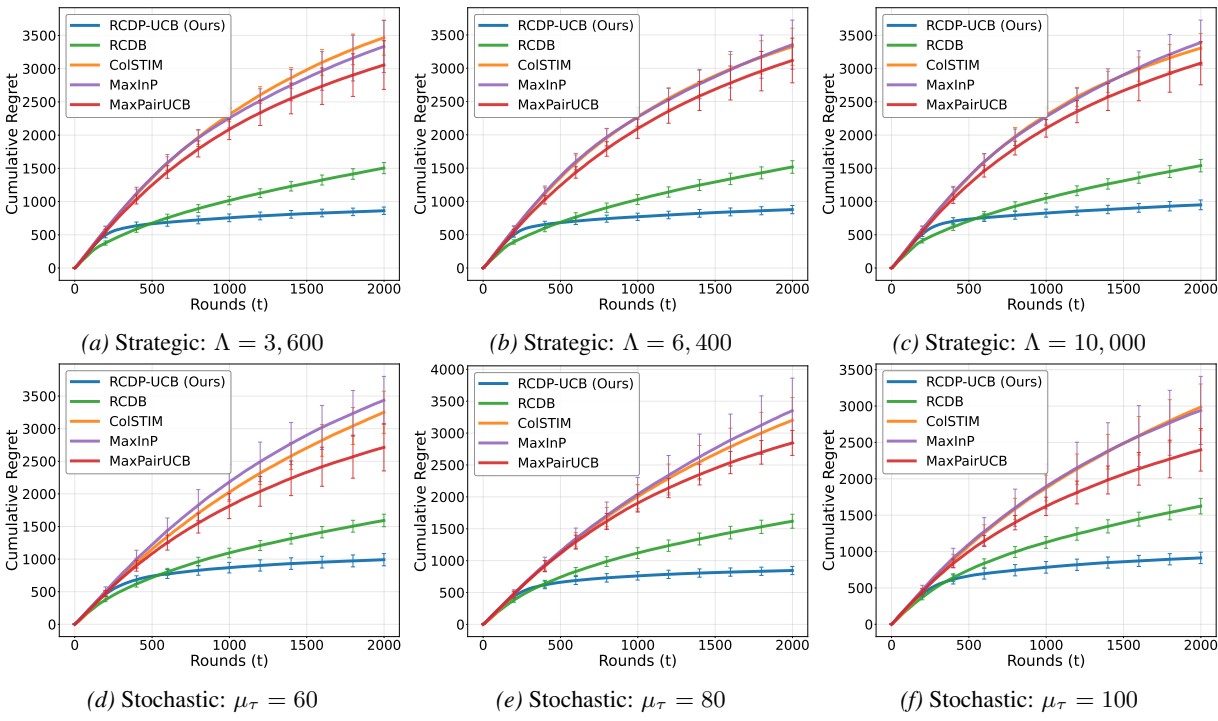

*Figure E.11.* **Polynomial Mapping (Pre-serving Baselines)**: Cumulative regret with varying delay budget $\Lambda$ and mean delay $\mu_\tau$ under adversarial corruption ($\mathcal{C} = 25$). All experiments were conducted across 10 runs with random seeds.

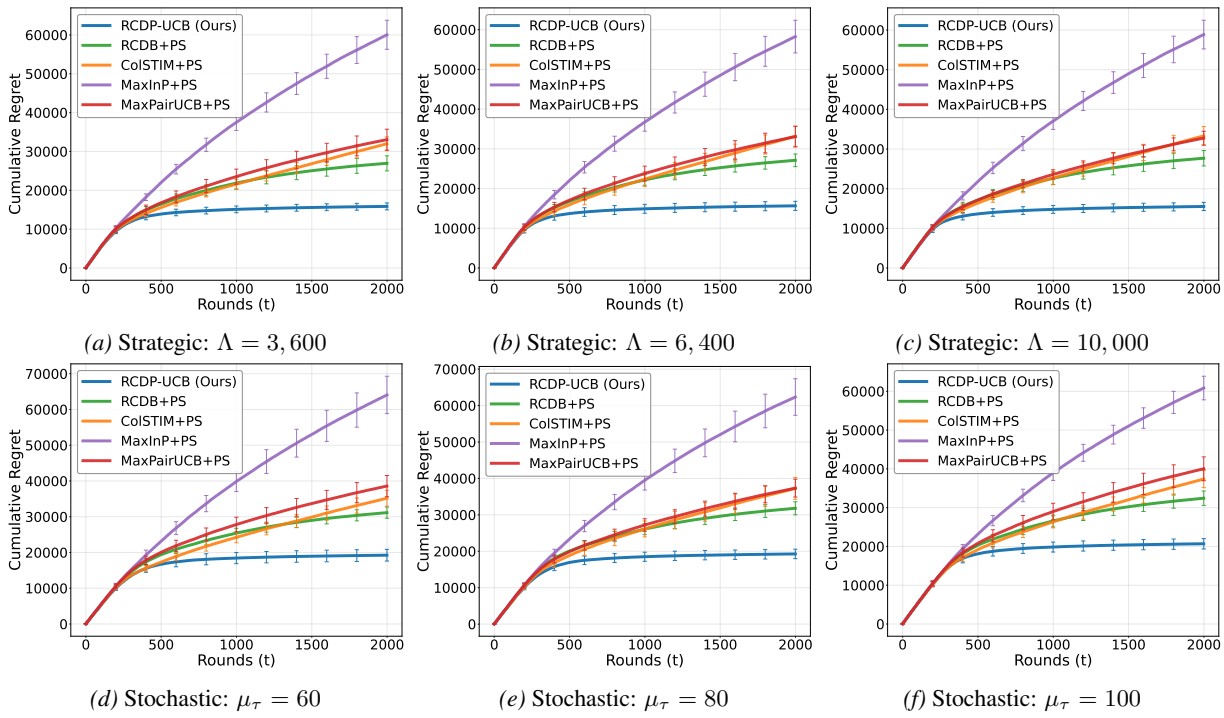

*Figure E.12.* **Absolute Mapping (Post-serving Learning)**: Cumulative regret with varying delay budget $\Lambda$ and mean delay $\mu_\tau$ under adversarial corruption ($\mathcal{C} = 25$, except $\Lambda = 10,000$ using $\mathcal{C} = 20$). All experiments were conducted across 10 runs with random seeds.

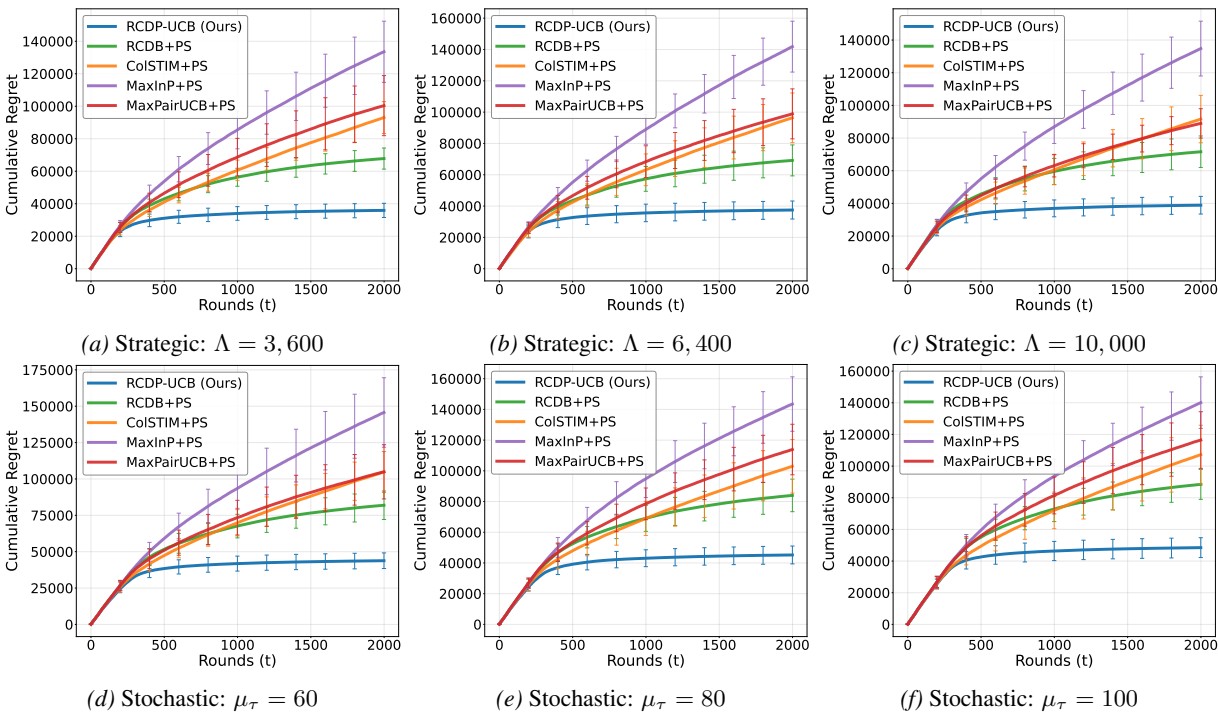

*Figure E.13.* **Polynomial Mapping (Post-serving Learning)**: Cumulative regret with varying delay budget $\Lambda$ and mean delay $\mu_\tau$ under adversarial corruption ($\mathcal{C} = 25$, except $\Lambda = 10,000$ using $\mathcal{C} = 20$). All experiments were conducted across 10 runs with random seeds.

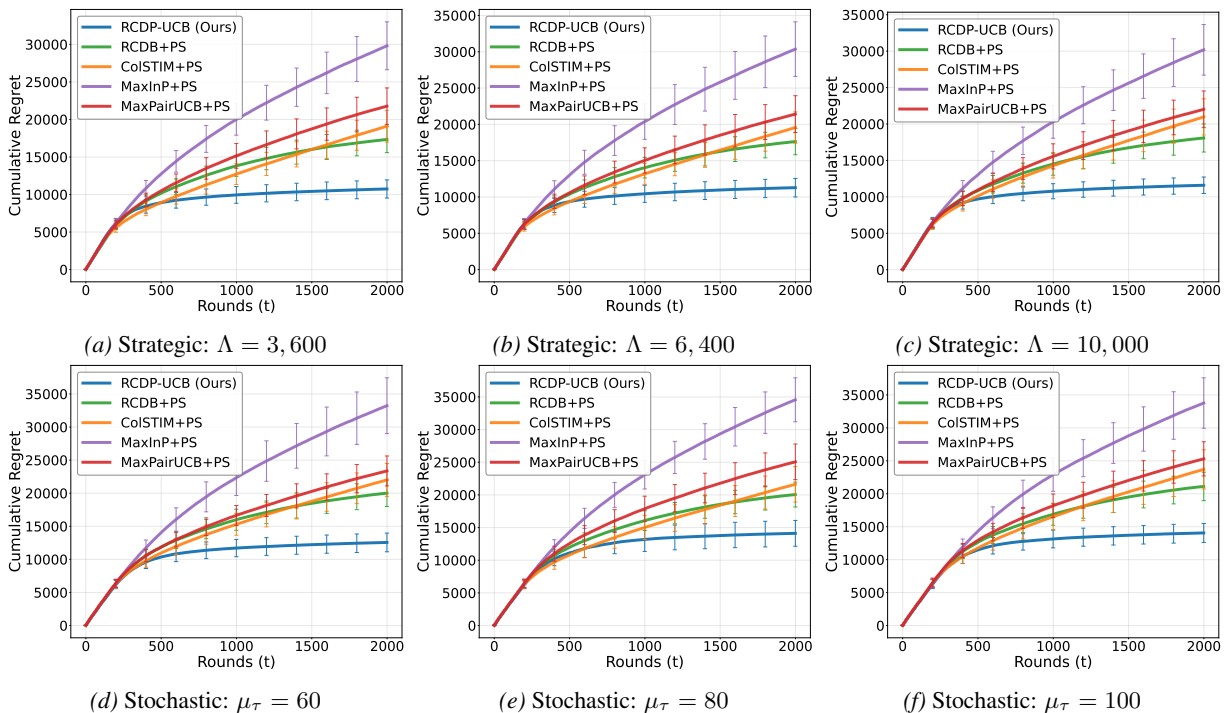

*Figure E.14.* **Sinusoidal Mapping (Post-serving Learning)**: Cumulative regret with varying delay budget $\Lambda$ and mean delay $\mu_\tau$ under adversarial corruption ($\mathcal{C} = 25$). All experiments were conducted across 10 runs with random seeds.

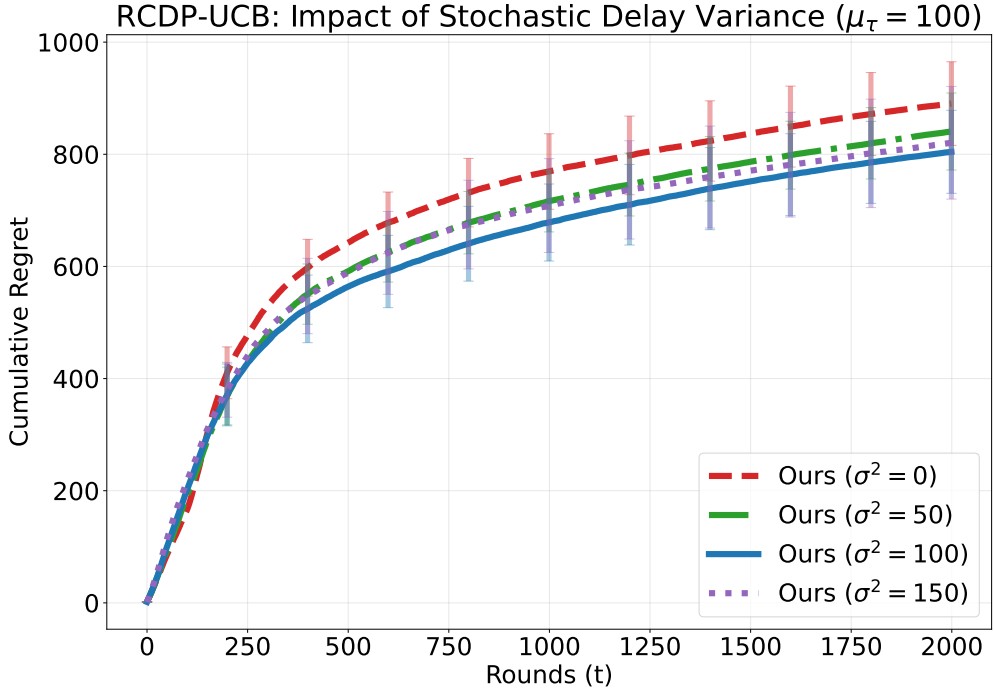

*Figure E.15.* **Stochastic Delay Variance Ablation**: Cumulative regret of RCDP-UCB under varying variances $\sigma^2 \in \{0, 50, 100, 150\}$ with fixed $\mu_\tau = 100$ and $\mathcal{C} = 25$. The results are aggregated over 10 runs.

