# OpenReview forum: "Robust Linear Dueling Bandits with Post-serving Context under Unknown Delays and Adversarial Corruptions"
_ICML.cc/2026/Conference — ICML 2026 regular_

### Official Review · Reviewer_XQuF · 2026-02-20

**Soundness:** 3
**Presentation:** 2
**Significance:** 2
**Originality:** 2
**Overall Recommendation:** 4
**Confidence:** 4

**Summary:**

This paper studies the challenging problem of linear contextual dueling bandits in the presence of pre-serving and post-serving contexts, unknown delays, and adversarial corruption in the observations.
Rather than directly applying previous approaches, which leads to an undesirable multiplicative dependence on the corruption $C$ and delay $D$, the authors introduce a separated weighting mechanism.
Specifically, they distinguish the use of between a full-information based weighting matrix $\tilde{V}_t$ and a delay-dependent matrix $\tilde{W}_t$.
Using this approach, they establish an estimation error bound for the true parameter with an additive dependence on both the corruption $C$ and the delay $D$.
This result yields $T$-optimal regret bounds when $\nu=1/2$, including the case where a linear mapping exists between pre-serving and post-serving contexts.

**Compliance With Llm Reviewing Policy:**

Affirmed.

**Final Justification:**

Since the authors have clearly addressed my concerns regarding the proofs, particularly the issues caused by typos in subscripts, I increase my score from 2 to 4.

**Key Questions For Authors:**

Q1) What happens if some components of the post-serving context are independent of the pre-serving context? How would this affect the proposed method and its theoretical guarantees?

Q2) The algorithm requires knowledge of $\kappa$, a lower bound on the derivative of the link function. While the existence of such a $\kappa$ is reasonable under standard assumptions, it is less clear whether assuming prior knowledge of its value is practical. Could the authors comment on this requirement?

Q3) My main concern is on the correctness of the theoretical guarantees, particularly the issues in comments 3-a) and 3-b).

**Limitations:**

yes

**Strengths And Weaknesses:**

### 1. Presentation

Overall, the paper is well-written and easy to follow. Below are some comments that may improve the clarity and readability.

1-a) The notation $d_u$ in L143 (p.3) is not defined.

1-b) The distinction between the corrupted observation $\gamma_t$ and the observation $o_t$ is unclear. Based on my understanding, $\gamma_t$ denotes the feedback that would be observed at round $s$ based on the true feedback $l_t$ if there were no delay, while $o_t$ denotes the actual observation received at round $t$ in the presence of delay (i.e., it could be $\gamma_{s'}$ for some $s'\leq s$). Under this interpretation, the statement "unknown whether $o_t = l_t$ or not" becomes confusing in the delayed feedback setting. A clearer explanation of the relationship among $l_t, \gamma_t, o_t$ would be helpful.

1-c) The sets $\mathcal{X}$ and $\mathcal{Y}$ are not explicitly defined.

1-d) Missing hat in L221 (RHS): $\Delta z$ should be $\Delta \hat{z}$.

1-e) The referencing style for equations is inconsistent. The authors use both "Equation (1)" and "Eq. (1)". Please unify the style.

1-f) In (A.5), the authors explicitly use a result from Bengs et al. (2022) in L245, which holds when the link function $g$ belongs to the exponential family (as far as I see the explanation). This suggests that Lemma 5.1 may require an additional assumption on $g$ beyond Assumption 4.1. It would be helpful to clarify whether the assumptions in Bengs et al. (2022) are necessary to imply Assumption 4.1. If not, the additional requirement should be explicitly stated as a separate assumption.

1-g) In L317, $c_s$ is used to denote whether the observation is corrupted or not, but this notation is already used for the exploration parameter. To avoid confusion, it would be better to use $C_s$ as used in the appendix.

1-h) In Remark 5.4, the authors state "by setting $\nu=1/2$", which suggests that $\nu \in (0, 1/2]$ can be chosen arbitrarily. However, Remark 4.1 indicates that the optimal choice of $\nu$ depends on the mapping between pre- and post-serving contexts. It would be better to clarify this point. For example, $\nu=1/2$ can be achievable when the context mapping is parametric linear or belongs to an RKHS.

1-i) There are several issues with the citation style. The formatting is inconsistent and in some cases incorrect:
* Di et al. (2023) should likely be Di et al. (2024) if referring to their ICLR 2024 paper (or specify arXiv 2023 if appropriate).
* Lancewicki et al. (2023) appears to list the authors in the wrong order. In the COLT2023 version, the first author is Dirk van der Hoeven and in the JMLR 2025 version, it is Luka Zierahn.
* Wang et al. (2024) should be Wang et al. (2023), as the paper was published at NeurIPS 2023.

---

### 2. Significance and Originality

Regarding the generality of the problem formulation, this work may be of interest to researchers in the bandit community. However, I am not fully convinced about its broader applicability.

In particular, it is unclear what happens if some components of the post-serving context are independent of the pre-serving context. Such a setting seems closer to the motivating examples provided in the Introduction (at least from my perspective). It would be helpful to clarify whether the proposed approach still applies in this case, and if so, how the guarantees would change.

Moreover, the assumption that post-serving contexts are observed immediately after selection appears somewhat unnatural in a delayed feedback scenario. In many practical settings, both the post-serving contexts and the feedback may only become observable after some delay, potentially at the same round. Under such a model, the key idea of leveraging $V_t$ instead of the delay-dependent $W_t$ would no longer be applicable. This would make the problem substantially more challenging, and it is possible that a multiplicative $CD$ dependence might be unavoidable in such cases.

To my understanding, this work represents one of the first attempts to study a more realistic and challenging variant of the problem. From this perspective, the above limitations may be understandable and acceptable, although discussing them explicitly would further strengthen the paper.

---

### 3. Soundness

There are two main issues in the proof of Theorem 5.2.

3-a) In Step 1 (p.15), the authors state that $r_t$ is written in terms of $\Delta z^\*$ rather than $\Delta z$.
However, the instantaneous regret defined in Section 2 is expressed in terms of $z$, not $z^\*$, and therefore includes the arm-dependent noise $\epsilon_t$. The definition with $z$ is natural since the best choice at round $t$ depends on the observed contexts $z_t$ rather than $z_t^\*$.

If the authors intend to maintain the current formulation, the regret expression should also account for the difference in noise terms between $k_t^\*, a_t, b_t$. While expressing $r_t$ in terms of $z^\*$ is valid in expectation, the analysis here focuses on high-probability bounds rather than expectations. Therefore, the two formulations are not equivalent, and this discrepancy needs to be addressed or clarified.

3-b) (B.11) and (B.12) appear to be incorrect. According to equation (5), $a_t$ and $b_t$ are maximizers with respect to $\Theta_{t-1}$, not $\Theta_t$. Therefore, (B.11) and (B.12) are not valid.

Moreover, since $\Theta_t$ is updated after selecting $a_t$ and $b_t$, the claimed relationships do not seem to follow without additional arguments or guarantees. This part of the proof requires further justification.

---
Below are minor comments.

3-c) A factor of 2 is missing in Lemma A.1 after applying the determinant–trace inequality.

3-d) Since $\Lambda$ denotes the budget, the first equality in L611 should be $\geq$ instead of $=$.

3-e) In Lemma A.2, under adversarial delays, the mean of sub-Gaussian delays may not exist. It would be better to explicitly specify the setting in which $\mu_\tau$ exists.

3-f) In Step 2 on p.13, is $\gamma_s$ a typo for $\zeta_s$?

---

> ### Author Rebuttal · Authors · 2026-03-25
>
> We appreciate the reviewer for the detailed reading. We address each concern below, beginning with the soundness questions (Q3: 3-a and 3-b), then the significance questions (Q1, Q2).
>
> ---
>
> ## 1. Soundness (Q3): 3-a and 3-b Are Notation-Level Typos
>
> **Both issues are notation mismatches between the algorithm/proof and the main text; all proofs and bounds are unaffected. We apologize for the confusion and fully address them as follows.**
>
> **[3-a] $z$ vs. $z^\ast$ in Regret Definition.**
>
> ***The proof (Appendix B) already uses $z^\ast\_{t,k}$ throughout.*** The regret *definition* in Section 3 should write  $z^\ast\_{t,k}$ instead of $z\_{t,k}$, which is the standard average regret. In our noisy setting, the oracle uses $z^\ast\_{t,k} = (x\_{t,k}, \phi\_\ast(x\_{t,k}))$, making the oracle's per-round value **deterministic**. The reviewer notes this is "valid in expectation" but not for high-probability bounds. However, defining regret over expected utilities rather than noisy realizations is standard in the dueling bandit literature (Saha, 2021; Bengs et al., 2022; Di et al., 2024). Since $z^\ast\_{t,k}$ is deterministic, there is no approximation gap. Fix: $z \to z^\ast$ in Section 3. ***Proof requires zero changes.***
>
> **[3-b] $\Theta\_t$ vs. $\Theta\_{t-1}$ in (B.11)–(B.12).**
> The algorithm (Eqs. 5–6) already uses $\Theta_{t-1}$. The appendix writes $\Theta_t$, a subscript typo.
>
> The reviewer correctly points out that (B.11)–(B.12) would not hold with $\Theta_t$. The algorithm (Eqs. 5–6) selects arms using $\Theta_{t-1}$, which is available at round $t$; with this correction, (B.11)–(B.12) hold by the optimality conditions of $a_t$ and $b_t$. The final bound results are completely unchanged.
>
>
>
> ---
>
> ## 2. Key Questions
>
> **Q1 (Independence of post-serving components):**
> If some post-serving components are independent of pre-serving contexts, $\phi\_\ast$ cannot meaningfully predict those components from $x$. In this case, **Assumption 4.3 (Learnability) cannot hold for those dimensions**: the approximation error $\|\hat{\phi}\_t(x) - \phi\_\ast(x)\|$ does not vanish in the independent subspace. Our guarantees thus apply to the **learnable subspace** where $t^{-\nu}$ convergence of $\hat{\phi}\_t$ is achievable. In practice, this is exactly the structure that arises: delivery time depends on restaurant location; food temperature depends on distance—precisely the components $x$ carries information about.
>
> **Q2 ($\kappa$):**
> Requiring knowledge of $\kappa$ is standard across all GLB literature (Filippi et al., 2010; Bengs et al., 2022). Since $g$ is a modeling choice made by the practitioner, $\kappa$ is an analytic property directly computable from the chosen $g$ (e.g., $\kappa=1/4$ for logistic). In particular, $\kappa$ need not be estimated from data—it is a closed-form constant derived from $g$, so knowing $\kappa$ requires no estimation once the practitioner specifies $g$.
>
> **Post-serving observed immediately:**
> The temporal separation is inherent to the application domain: post-serving contexts (e.g., delivery time, food temperature) are physically observed at service time, while *preference feedback* (e.g., user rating) arrives with delay. If all information—including post-serving contexts—were delayed, $\widetilde{V}\_t$ could not be built from the full history, and multiplicative $O(\mathcal{CD})$ coupling would be unavoidable. We have identified this boundary as an open problem in Section 7 (Limitations). Our algorithm exploits this temporal structure, and we view extending to fully delayed contexts as a promising future direction.
>
> ---
>
> ## 3. Minor Comments
> All minor comments (1-a through 1-i, 3-c through 3-f) are well taken and will be incorporated. In particular: the exponential family condition on $g$ (1-f) is already used in our proofs and will be explicitly stated as a formal assumption; the sub-Gaussian delay statement (3-e) will be clarified to specify that $\mu\_\tau$ applies only to the stochastic delay case. We will also improve notation consistency across the paper for readability.
>
> ---
>
> **We hope we have demonstrated that the soundness concerns are notation-level typos not affecting any proof or bound.** If these clarifications fully address the concerns, we would welcome a re-evaluation.
>
> ---
>
> ## References
>
> - Saha, Aadirupa. "Optimal algorithms for stochastic contextual preference bandits." Advances in Neural Information Processing Systems 34 (2021): 30050-30062.
> - Bengs, Viktor, Aadirupa Saha, and Eyke Hüllermeier. "Stochastic contextual dueling bandits under linear stochastic transitivity models." International Conference on Machine Learning. PMLR, 2022.
> - Filippi, Sarah, et al. "Parametric bandits: The generalized linear case." Advances in neural information processing systems 23 (2010).

---

> > ### Author Rebuttal · Reviewer_XQuF · 2026-04-01
> >
> > I appreciate the authors for correction of previous notation!
> >
> > Just to confirm, in that case, $\Theta_t$ in Lemma 5.1 is also $\Theta_{t-1}$ (or $\tilde{V}\_{t-1} -> \tilde{V}\_t$)?
> > If so, I am not sure how to modify (A.8) at a glance as the adaptive weights at round $s$ are defined w.r.t. $V$ at round $s-1$.
> > I ask this since Lemma 5.1 is used in (B.16) for the term $\Vert \Theta\_\* - \Theta\_{t-1}  \Vert\_{\tilde{V}\_{t-1}}$ (after modifying the subscript).
> >
> > Best regards,
> > Reviewer XQuF

---

> > > ### Author Response · Authors · 2026-04-02
> > >
> > > We thank the reviewer for the careful follow-up and appreciate the opportunity to clarify.
> > >
> > > ---
> > >
> > > **Yes.** As stated in our previous response regarding [3-b], $\Theta\_t$ should be corrected to $\Theta\_{t-1}$ throughout the proofs and lemma/theorem statements, including Lemma 5.1. The object labeled $\Theta\_t$ in Appendix A is the estimator built from data $\{1,\ldots,t{-}1\}$ whose feedback has arrived by round $t$, and renaming it $\Theta\_{t-1}$ aligns with the algorithm (Eqs. 5–6). The design matrices $\widetilde{V}\_{t-1}$ and $\widetilde{W}\_{t-1}$ retain their subscripts — after the correction, $||\Theta\_{t-1} - \Theta\_\ast||\_{\widetilde{V}\_{t-1}}$ in Lemma 5.1 is bounded independently of $t$, where all subscripts are $t{-}1$, consistently referring to data from rounds $1,\ldots,t{-}1$. This is consistent with the standard convention used in the literature. For instance, Lemma 5.1 in Di et al. (2025, ICML — RCDB) showed that $||\theta\_t - \theta^\ast||\_{\Sigma\_t}$ is bounded with both $\theta\_t$ and $\Sigma\_t$ built from data $1,\ldots,t{-}1$, i.e., **the estimator and its associated design matrix share the same temporal index**. We acknowledge that this subscript inconsistency — though not affecting any derivation — can cause confusion for the reader, and we will standardize the notation throughout the revised manuscript.
> > >
> > > We now explain why (A.8) and (B.16) remain valid after this correction, and provide a detailed walkthrough.
> > >
> > > ---
> > >
> > > **Why (A.8) is unaffected.** The adaptive weight $\omega\_s = \min \left(1, \alpha / ||\Delta z\_s||\_{\widetilde{V}\_{s-1}^{-1}}\right)$ is computed at action time $s$ using $\widetilde{V}\_{s-1}$, the covariate matrix up to round $s{-}1$. This definition depends only on $\widetilde{V}\_{s-1}$, not on the subscript of $\Theta$. In the proof, the key step that invokes the clipping property is: for every $s \leq t{-}1$,
> > > $$\omega_s ||\Delta z\_s||\_{\widetilde{V}\_{t-1}^{-1}} \leq \omega\_s ||\Delta z\_s||\_{\widetilde{V}\_{s-1}^{-1}} \leq \alpha,$$
> > > where the first inequality follows from $\widetilde{V}\_{s-1}^{-1} \succeq \widetilde{V}\_{t-1}^{-1}$ (monotonicity of the design matrix) for $t\geq s$, and the second follows directly from $\min(x,y) \leq y$, i.e., $\omega\_s \leq \alpha / ||\Delta z\_s||\_{\widetilde{V}\_{s-1}^{-1}}$. This chain is entirely independent of the $\Theta$ subscript.
> > >
> > > ---
> > >
> > > **Why (B.16) is unaffected.** In the regret decomposition (Appendix B), the estimation error term $\mathcal{B}\_{3,t}$ is:
> > > $$\mathcal{B}\_{3,t} = \langle \Theta\_\ast - \Theta\_{t-1}, \Delta\widehat{z}\_{t,k\_t^\ast,a\_t} + \Delta\widehat{z}\_{t,k\_t^\ast,b\_t}\rangle.$$
> > > By the Cauchy–Schwarz inequality and (B.13),
> > > $$\mathcal{B}\_{3,t} \leq ||\Theta\_\ast - \Theta\_{t-1}||\_{\widetilde{V}\_{t-1}} \left(2||\Delta\widehat{z}\_{t,k\_t^\ast,a\_t}||\_{\widetilde{V}_{t-1}^{-1}} + ||\Delta\widehat{z}\_{t,a_t,b_t}||\_{\widetilde{V}\_{t-1}^{-1}}\right).$$
> > > The corrected Lemma 5.1 gives that $||\Theta\_{t-1} - \Theta\_\ast||\_{\widetilde{V}\_{t-1}}$ is bounded independently of $t$, which directly bounds the first factor. We acknowledge that the original (B.16) did not make this application of Lemma 5.1 sufficiently explicit, and we will expand this step in the revision to clarify the connection.
> > >
> > > ---
> > >
> > > **In summary,** the correction $\Theta\_t \to \Theta\_{t-1}$ propagates cleanly through the entire proof chain — the concentration lemma (Appendix A), the adaptive weight bounds, and the regret decomposition (Appendix B). All intermediate inequalities remain valid because (i) the weights $\omega\_s$ are defined w.r.t. $\widetilde{V}\_{s-1}$, independent of the $\Theta$ subscript, and (ii) the design matrices satisfy $\widetilde{V}\_{t-1} \succeq \widetilde{V}\_{s-1}$ for all $t \geq s$. We are grateful for the reviewer's careful reading, which has helped us identify notation that, while not affecting the proofs, could mislead readers. We will thoroughly standardize the subscripts in the revision to eliminate any ambiguity.
> > >
> > > ---
> > >
> > > **We hope this fully addresses the reviewer's remaining concerns.**
> > >
> > > ---
> > >
> > > # References
> > >
> > > Di, Qiwei, Jiafan He, and Quanquan Gu. "Nearly Optimal Algorithms for Contextual Dueling Bandits from Adversarial Feedback." International Conference on Machine Learning. PMLR, 2025.

---

### Official Review · Reviewer_cumU · 2026-03-12

**Soundness:** 4
**Presentation:** 4
**Significance:** 3
**Originality:** 3
**Overall Recommendation:** 5
**Confidence:** 4

**Summary:**

The paper presents an algorithmic framework called RCDP-UCB (Robust to Corruption, Delay, and Post-serving UCB) for dueling linear bandits with post-serving contexts, adversarial corruptions, and delays. The core contribution of the paper is in an algorithm and its complexity analysis which ensures that delay and adversarial corruption enter additively in the regret as opposed to being multiplicative that is obtained by naively extending prior analysis with one of the two non-idealities in the reward feedback model.

**Compliance With Llm Reviewing Policy:**

Affirmed.

**Key Questions For Authors:**

N/A

**Limitations:**

Yes

**Strengths And Weaknesses:**

The paper is extremely cleanly written! I must commend the authors for such a clean write-up. Not just the main paper, even the proofs are extremely cleanly presented. I went through most of the appendix almost line-by-line and the logic/derivation is really lucid!

The core idea of the algorithm design and the proof is actually quite simple: the context information comes without delay, even when the reward observation can be delayed. I would urge the authors to think a bit more carefully why this distinction translates to the additive regret contributions than multiplicative. The intuition was not clear to me. However, given the lucid nature of their proofs, I imagine that it can be cleanly seen from the regret decomposition somehow. That point might be worthwhile mentioning in the main body of the paper.

Contribution-wise, the paper does provide a one-stop shop for the state-of-the-art analysis of contextual dueling linear bandits. While they do not innovate a whole lot in the non-idealities they consider, the analysis of all of it under one roof is a good contribution.

---

> ### Author Rebuttal · Authors · 2026-03-25
>
> We thank the reviewer for the generous assessment and the line-by-line verification of the appendix.
>
> ---
>
> ## 1. Intuition: Why Additive, Not Multiplicative?
>
> The reviewer's observation is precisely correct: **context information arrives without delay, while only preference feedback is delayed.** We elaborate on *why* this yields the additive regret.
>
> **The standard approach and its failure.**
> Prior delayed feedback methods construct $\widetilde{W}\_t = \lambda I + \kappa \sum\_{s: s+\tau\_s < t} \omega\_s \Delta z\_s \Delta z\_s^\top$ from *arrived feedback only*. Delay shrinks $\widetilde{W}\_t$, inflating $\widetilde{W}\_t^{-1}$. Since corruption bias is measured through this inflated inverse, each corrupted sample's error is amplified by missing data — yielding $O(\mathcal{C} \cdot \mathcal{D})$ multiplicative coupling. Combining existing methods (Di et al., 2024; Howson et al., 2023) cannot escape this; the coupling is structural.
>
> **Our key insight: separate what you *know* from what you *learn*.**
> Contexts (both pre-serving $x\_{t,k}$ and post-serving $y\_{t,k}$) are observed **immediately** at round $t$, while preference feedback $o\_t$ arrives with adversarial delay $\tau\_t$. The learner *knows* the full context $z\_{t,k}$ at round $t$ even though the reward has not arrived. We exploit this by constructing $\widetilde{V}\_t = \lambda I + \kappa \sum\_{s=1}^{t} \omega\_s \Delta z\_s \Delta z\_s^\top$ from the **entire** context history. This matrix is **delay-invariant**: independent of $\{\tau\_t\}$.
>
> However, the MLE $\Theta\_t$ uses *arrived* feedback only. So the algorithm uses $\widetilde{V}\_t$ for **arm selection** (UCB and robustness weighting) while $\Theta\_t$ is updated only from arrived feedback. This deliberate mismatch — a richer matrix for decisions than for estimation — is the source of the additive structure.
>
> **Why this directly yields additive regret.**
> The cleanest way to see it is from the three-way partition of history (Lemma 5.1):
> $$[t-1] = (\mathcal{A}\_t \cap \mathcal{E}^c) \;\sqcup\; (\mathcal{A}\_t \cap \mathcal{E}) \;\sqcup\; \mathcal{A}\_t^c$$
> i.e., (arrived & clean) $\sqcup$ (arrived & corrupted) $\sqcup$ (pending).
>
> Under $\widetilde{V}\_t$, each set contributes to the confidence width **independently**:
>
> - **Arrived & clean** $\to$ standard martingale noise $\to$ $O(\sqrt{d \log T})$.
> - **Arrived & corrupted** $\to$ bounded by $O(\mathcal{C}\alpha)$, measured against the full, non-inflated $\widetilde{V}\_t$.
> - **Pending** $\to$ bounded by $O(\mathcal{D}\alpha)$; these contexts already stabilize $\widetilde{V}\_t$, so "missing feedback" does not shrink the matrix.
>
> Since corruption and delay each contribute $O(\alpha)$ bias in the *same norm* against a *non-shrunk* matrix, they decouple into additive terms. Setting $\alpha = \sqrt{d}/(\mathcal{C}+\mathcal{D})$ balances bias against variance, yielding confidence radius $\beta\_t = \widetilde{O}(\sqrt{d})$ and regret $\widetilde{O}(d(\sqrt{T} + \mathcal{C} + \mathcal{D}))$. Our lower bound $\Omega(\sqrt{dT} + d\mathcal{C} + \mathcal{D}')$ (Theorem 5.2) confirms that this additive cost structure is information-theoretically fundamental — not an artifact of our proof technique.
>
> **The role of post-serving contexts — and why their timing matters.**
> If *all* information — including post-serving contexts — were delayed alongside the preference feedback, the learner would have no choice but to build the design matrix from arrived data only, making $O(\mathcal{C} \cdot \mathcal{D})$ multiplicative coupling unavoidable. However, in realistic post-serving settings, contexts such as delivery time and food temperature are physically observed at service time, *before* the delayed feedback. Our algorithm exploits exactly this: since both $x\_{t,k}$ and $y\_{t,k}$ are available at round $t$, $\widetilde{V}\_t$ incorporates $\Delta z\_s$ from *all* past rounds without waiting for feedback. This delay-invariant structure is what makes the additive bound possible.
>
> **We will revise a paragraph in Section 5 for presenting this intuition clearly, as the reviewer suggests.** The regret decomposition is indeed the right lens to see why immediate context and delayed reward translates to the additive structure. We welcome any further questions.
>
> ---
>
> ## References
>
> - Abbasi-Yadkori, Yasin, Dávid Pál, and Csaba Szepesvári. "Improved algorithms for linear stochastic bandits." Advances in neural information processing systems 24 (2011).
> - Di, Qiwei, Jiafan He, and Quanquan Gu. "Nearly optimal algorithms for contextual dueling bandits from adversarial feedback." arXiv preprint arXiv:2404.10776 (2024).
> - Howson, Benjamin, Ciara Pike-Burke, and Sarah Filippi. "Delayed feedback in generalised linear bandits revisited." International Conference on Artificial Intelligence and Statistics. PMLR, 2023.

---

> > ### Author Rebuttal · Reviewer_cumU · 2026-04-01
> >
> > I enjoyed reading the explanation based on the three-way partition of the algorithm's history. Adding the intuition to the main body of the paper will be a nice addition.

---

> > > ### Author Response · Authors · 2026-04-02
> > >
> > > We thank the reviewer for the positive feedback. We will incorporate the three-way partition intuition into the main body of the revised manuscript as suggested.
> > >
> > >
> > > ---
> > >
> > > Dear Reviewer,
> > >
> > > As the deadline for final justification is April 7 AOE,
> > > we would kindly ask you to finalize the reassessment scores at your earliest convenience.

---

### Official Review · Reviewer_xzUj · 2026-03-13

**Soundness:** 3
**Presentation:** 3
**Significance:** 3
**Originality:** 3
**Overall Recommendation:** 4
**Confidence:** 3

**Summary:**

This paper studies linear contextual dueling bandits with three complications combined: post-serving contexts, delayed feedback, and adversarial corruption. The authors propose RCDP-UCB algorithm, which achieves a regret bound of $\tilde{O(d(\sqrt{T}+C+D)), where $C$ is the corruption level, $D$ is a delay related parameter. The paper also gives lower bound without post-serving and includes experiments on synthetic dataset.

**Compliance With Llm Reviewing Policy:**

Affirmed.

**Final Justification:**

This is a technically solid paper, and therefore, I remain positive about the paper.

**Key Questions For Authors:**

See weakness.

**Limitations:**

Yes.

**Strengths And Weaknesses:**

Strength

+The problem is well-motivated and the combination of post-serving contexts, delay, and corruption in dueling bandits is a natural problem setting to consider

+The idea of the algorithm is clearly illustrated: add robust weights over design matrices and use $\alpha$ parameter to simultaneously handle the delay and the corruption.

+The theoretical result is complete with both upper bounds and lower bounds. The summary table is also a plus for the presentation.

+The experiments validate the theoretical claims over synthetic dataset.

Weakness

-While the combination is a natural direction in the literature, the core algorithmic rule seems quite close to existing robust linear/dueling bandits, and much of the contribution is to choose the second arm through the uncertainty term in Equation (6), together with a specific weighting procedure to handle delay and corruptions.

-The empirical section is weaker than the theory section. All results shown are synthetic, the number of runs is small, and the baselines are adapted in a way that may not be strong for the post-serving setting.

---

> ### Author Rebuttal · Authors · 2026-03-25
>
> We thank the reviewer for the positive assessment. We address the two weaknesses below.
>
> ---
>
> ## 1. Technical Novelty
>
> We respectfully disagree that the contribution reduces to Eq. (6) plus a weighting procedure. The simplicity is a consequence of novelty — not evidence against it.
>
> **The core problem.** Every prior method builds $\widetilde{W}\_t$ from *arrived feedback only*. When delay and corruption coexist, delay inflates $\widetilde{W}\_t^{-1}$; corruption bias is amplified through this inflated inverse, yielding $O(\mathcal{CD})$ multiplicative regret. This coupling is structural to the $\widetilde{W}\_t$ paradigm.
>
> **Our solution: changing arm selection itself.** We select arms using $\widetilde{V}\_{t-1}$, built from the *entire* context history, instead of $\widetilde{W}\_{t-1}$. This redefines *what information the agent uses to choose arms*.
>
> **Why this idea is counterintuitive.** Every existing algorithm selects arms based on data it has *received*. Using $\widetilde{V}\_{t-1}$ means selecting arms with a matrix that includes contexts whose *rewards have not yet arrived* — seemingly paradoxical. The insight is that contexts ($x\_{s,a}$, $z\_{s,a}$) are observed *immediately* when an arm is played; only the reward is delayed. Building $\widetilde{V}\_{t-1}$ from all observed contexts yields a matrix *invariant to delay*: its growth depends on $t$, not on arrived rewards. This eliminates the delay-corruption coupling. $\widetilde{V}\_{t-1}^{-1}$ does not inflate with delay, so corruption bias is not amplified. A single $\alpha = \sqrt{d}/(\mathcal{C}+\mathcal{D})$ handles both phenomena, whereas prior methods required separate, conflicting mechanisms.
>
> **The proof is also non-trivial.** This shift breaks three proof techniques: (i) confidence must hold w.r.t. $\widetilde{V}\_{t-1}$ including pending rounds; (ii) adversarial delays destroy the martingale structure; (iii) corruption and delay unify under $\widetilde{V}\_t$ but conflict under $\widetilde{W}\_t$. Resolving these yields the first additive $O(\mathcal{C}+\mathcal{D})$ bound with matching lower bounds. **We will revise Section 5 to make this explicit.**
>
> ---
>
> ## 2. Empirical Section
>
> We added **real-world experiments** on two recommendation datasets following Wang et al. (NeurIPS 2023).
>
> **Setup.** **MovieLens-100K** (sparse) and **Jester** (dense). $d{=}5$, $e{=}3$, $K{=}10$, $T{=}2{,}000$, 5 runs. $\mathcal{C} = \Lambda = 49$, stochastic delay $\mu\_\tau{=}7, \sigma\_\tau{=}2$.
>
> **Table 1. Final cumulative regret at $T = 2{,}000$ (mean $\pm$ std, 5 runs).**
>
> | Method              | MovieLens (Strategic) | MovieLens (Stochastic) | Jester (Strategic)  | Jester (Stochastic) |
> | ------------------- | --------------------- | ---------------------- | ------------------- | ------------------- |
> | **RCDP-UCB (Ours)** | **5,895 $\pm$ 833**   | **5,881 $\pm$ 758**    | **6,107 $\pm$ 531** | **6,140 $\pm$ 468** |
> | RCDB+PS             | 10,656 $\pm$ 798      | 10,661 $\pm$ 840       | 10,836 $\pm$ 675    | 10,945 $\pm$ 662    |
> | ColSTIM+PS          | 11,518 $\pm$ 977      | 12,567 $\pm$ 1,717     | 14,788 $\pm$ 2,227  | 15,463 $\pm$ 3,136  |
> | MaxInP+PS           | 16,795 $\pm$ 2,719    | 15,624 $\pm$ 2,197     | 16,373 $\pm$ 1,705  | 16,219 $\pm$ 1,420  |
>
> RCDP-UCB achieves **1.8$\times$ lower regret** than RCDB+PS and **2.5–2.8$\times$ lower** than MaxInP+PS.
>
> **Table 2. Average per-round regret $R(t)/t$ for RCDP-UCB.**
>
> | $t$   | MovieLens (Strat.) | MovieLens (Stoch.) | Jester (Strat.) | Jester (Stoch.) |
> | ----- | ------------------ | ------------------ | --------------- | --------------- |
> | 500   | 5.33               | 5.40               | 5.30            | 5.37            |
> | 1,000 | 3.93               | 3.97               | 3.91            | 3.96            |
> | 2,000 | **2.95**           | **2.94**           | **3.05**        | **3.07**        |
>
> Decreasing $R(t)/t$ confirms **sublinear regret** on real data. **(1)** RCDP-UCB generalizes across heterogeneous (MovieLens) and consensus-driven (Jester) domains; **(2)** regret is nearly identical under strategic vs. stochastic delays, confirming delay agnosticism; **(3)** all baselines share the same $\hat{\phi}\_t$ — the gap is purely algorithmic. We will include these in the appendix and increase to $\geq 10$ runs. If these address the concerns, we welcome a re-evaluation.
>
> ---
>
> We hope we have addressed all the reviewer's concerns.
>
> ---
>
> ## References
>
> - Di, Qiwei, Jiafan He, and Quanquan Gu. "Nearly optimal algorithms for contextual dueling bandits from adversarial feedback." arXiv preprint arXiv:2404.10776 (2024).
> - Howson, Benjamin, Ciara Pike-Burke, and Sarah Filippi. "Delayed feedback in generalised linear bandits revisited." International Conference on Artificial Intelligence and Statistics. PMLR, 2023.
> - Wang, Chaoqi, et al. "Follow-ups also matter: improving contextual bandits via post-serving contexts." Advances in Neural Information Processing Systems 36 (2023): 12774-12796.

---

> > ### Author Rebuttal · Reviewer_xzUj · 2026-04-02
> >
> > Thanks for the authors' rebuttal. I remain positive about the paper.

---

> > > ### Author Response · Authors · 2026-04-02
> > >
> > > We sincerely appreciate the reviewer's continued support and the constructive feedback throughout the review process.
> > >
> > >
> > > ---
> > >
> > > Dear Reviewer,
> > >
> > > As the deadline for final justification is April 7 AOE,
> > > we would kindly ask you to finalize the reassessment scores at your earliest convenience.

---

### Official Review · Reviewer_cqey · 2026-03-23

**Soundness:** 3
**Presentation:** 3
**Significance:** 3
**Originality:** 2
**Overall Recommendation:** 4
**Confidence:** 3

**Summary:**

This paper studies a linear dueling bandit problem under (1) post-serving context, (2) delayed feedback, and (3) adversarial corruption. Then an algorithm is developed, and its regret is analyzed as a function of problem parameters, including the delay complexity and the adversarial corruption. Then there is a lower bound analysis, and the result shows that the regret of the proposed algorithms is nearly optimal. Numerical experiments also show that the proposed algorithm outperforms multiple different baselines under different settings.

**Compliance With Llm Reviewing Policy:**

Affirmed.

**Key Questions For Authors:**

Can authors explain some special cases of their results when a subset of the model is relaxed to more basic versions? For example, how the problem becomes different if we keep 2 out of 3 modeling assumptions (post-serving context, delay, corruption) and relax the last one in the base setting?

**Strengths And Weaknesses:**

**Strengths**

-- The problem is well-motivated from a practical side.

-- The related work is very comprehansive and provides a clear positioning of the paper in this literature which is at the intersection of multiple lite of work.

-- The presented algorithm and its regret analysis make sense, and the results are clearly presented.

-- The results sound significant as compared to what is already in the literature and what the lower bound suggests.

-- The numerical results do a good job of providing a sufficient number of baselines and a sufficient number of settings to compare with. And it also shows the results are promising.


** Weaknesses**

-- While the results presented are very clear, it is not obvious where the core unique technical challenge of the proposed algorithm and the setting are. The paper considers a setting that has three unique dimensions (post-serving context, delayed reward, and corruption), and it is hard to see where exactly the core technical challenge is coming.

---

> ### Author Rebuttal · Authors · 2026-03-25
>
> We thank the reviewer for the positive assessment.
>
> ## 1. Where is the Core Technical Challenge?
>
> **The core challenge: delay and corruption couple multiplicatively.** No prior work in contextual (dueling) bandits jointly addresses these two. Every existing method builds $\widetilde{W}\_t$ from *arrived feedback only*. Delay inflates $\widetilde{W}\_t^{-1}$, and corruption bias is amplified through this inflated inverse, producing $O(\mathcal{CD})$. Combining Di et al. (2024) for corruption with Howson et al. (2023) for delay cannot escape this — the coupling is structural.
>
> **Our key insight: anchoring to the full information geometry.** We replace $\widetilde{W}\_t$ with $\widetilde{V}\_t$, built from the *entire* context history with adaptive weighting $\omega\_s = \min(1, \alpha/||\Delta z\_s||\_{\widetilde{V}\_{s-1}^{-1}})$. Since contexts are observed at action time (even when outcomes are delayed), $\widetilde{V}\_t$ is delay-invariant by construction. A single $\alpha = \sqrt{d}/(\mathcal{C}+\mathcal{D})$ simultaneously absorbs corruption ($O(\mathcal{C}\alpha)$) and delay ($O(\mathcal{D}\alpha)$), without requiring knowledge of the delay type. Prior approaches required *separate* mechanisms that conflict when combined.
>
> **Why this is non-trivial — three proof-level difficulties:**
>
> *(i) Estimator–matrix mismatch.* $\Theta\_{t-1}$ is computed from arrived data (via $\widetilde{W}\_t$), yet confidence must hold w.r.t. $\widetilde{V}\_{t-1}$ including pending rounds — violating self-normalized concentration, since $\widetilde{V}\_{t-1} \succ \widetilde{W}\_{t-1}$.
>
> *(ii) Broken martingale.* Under delays, $\eta\_s$ and $\mathbb{I}\_{s+\tau\_s \leq t}$ are dependent, collapsing the martingale structure. We decompose the error into a proper martingale minus a residual bounded by $O(\mathcal{D}\alpha)$ via norm decoupling.
>
> *(iii) Additive decomposition.* A naive analysis yields $O(\mathcal{C}\mathcal{D})$. We partition history into (arrived & clean) $\sqcup$ (arrived & corrupted) $\sqcup$ (pending), controlling each *independently*: corruption $O(\mathcal{C}\alpha)$, delay $O(\mathcal{D}\alpha)$, yielding **additive** $\widetilde{O}(\sqrt{d} + \mathcal{C}\alpha + \mathcal{D}\alpha)$.
>
> **The resolution.** Setting $\alpha = \sqrt{d}/(\mathcal{C}+\mathcal{D})$ gives $\beta\_t = \widetilde{O}(\sqrt{d})$ and regret $\widetilde{O}(d(\sqrt{T} + \mathcal{C} + \mathcal{D}))$. Our lower bound $\Omega(\sqrt{dT} + d\mathcal{C} + \mathcal{D}')$ confirms the additive structure is information-theoretically fundamental.
>
> **Post-serving adds a further challenge.** The learner faces a three-way feature mismatch: estimated $\widehat{z}\_{t,k}$ for selection, noisy $z\_{t,k}$ for weighting, and latent $z^\ast\_{t,k}$ for reward. Learning $\widehat{\phi}\_t$ itself relies on corrupted/delayed feedback. The same $\widetilde{V}\_t$-mechanism ensures the approximation error never interacts multiplicatively with $\mathcal{C}$ or $\mathcal{D}$.
>
> ## 2. Special Cases (Keeping 2 of 3 Assumptions)
>
> | Setting | Drop | Our Results | Difficulty |
> |---|---|---|---|
> | Corruption+Delay | Post-serving | $\widetilde{O}(d(\sqrt{T}+\mathcal{C}+\mathcal{D}))$ if $d=O(d\_x)$ | **Hard.** Requires full $\widetilde{V}\_t$ machinery. |
> | Corruption+Post-serving | Delay | $\widetilde{O}(d(\sqrt{T}+\mathcal{C})+\sqrt{dd\_x}\sqrt{T})$ | Moderate. Clipping alone suffices. |
> | Delay+Post-serving | Corruption | $\widetilde{O}(d(\sqrt{T}+\mathcal{D})+\sqrt{dd\_x}\sqrt{T})$ | Moderate. Standard decomposition. |
> | Standard | All three | $\widetilde{O}(d\sqrt{T})$ | Baseline. |
>
> **Corruption+delay is the hard case** — multiplicative coupling only arises when both are present. Rows 2–3 extend existing results, as $\widetilde{V}\_t$ degenerates to $\widetilde{W}\_t$ when delays are absent. The algorithm is also *delay-regime-agnostic*: the same $\alpha$ adapts to both regimes, with $\mathcal{D}$ scaling as $\mu\_\tau$ (sub-Gaussian) or $\Lambda^{1/2}$ (adversarial). **We will add this table in Section 5.**
>
>  We hope these clarifications address the reviewer's questions.
>
> ## References
>
> - Abbasi-Yadkori, Yasin, Dávid Pál, and Csaba Szepesvári. "Improved algorithms for linear stochastic bandits." Advances in neural information processing systems 24 (2011).
> - Bengs, Viktor, Aadirupa Saha, and Eyke Hüllermeier. "Stochastic contextual dueling bandits under linear stochastic transitivity models." International Conference on Machine Learning. PMLR, 2022.
> - Di, Qiwei, Jiafan He, and Quanquan Gu. "Nearly optimal algorithms for contextual dueling bandits from adversarial feedback." arXiv preprint arXiv:2404.10776 (2024).
> - Howson, Benjamin, Ciara Pike-Burke, and Sarah Filippi. "Delayed feedback in generalised linear bandits revisited." International Conference on Artificial Intelligence and Statistics. PMLR, 2023.
> - Saha, Aadirupa. "Optimal algorithms for stochastic contextual preference bandits." Advances in Neural Information Processing Systems 34 (2021): 30050-30062.

---

> > ### Author Rebuttal · Reviewer_cqey · 2026-04-01
> >
> > The authors have addressed my concerns.

---

> > > ### Author Response · Authors · 2026-04-02
> > >
> > > We are grateful for the reviewer's thorough evaluation and are glad that our responses have resolved the concerns.
> > >
> > > ---
> > >
> > > Dear Reviewer,
> > >
> > > As the deadline for final justification is April 7 AOE,
> > > we would kindly ask you to finalize the reassessment scores at your earliest convenience.

---

### Decision · Program_Chairs · 2026-04-30

**Decision:**

Accept (regular)

**Comment:**

This paper studies the problem of linear dueling bandits in an environment with adversarial corruptions, post-serving context and delays.

They derive an algorithm for this problem and analyse its regret. They provide almost matching lower bounds in the problem parameters and conduct empirical experiments.

All reviewers agree that this is a solid paper with a clear presentation, well-motivated problem and a sufficient technical contribution.